# Design and Synthesis of Novel Amino and Acetamidoaurones with Antimicrobial Activities

**DOI:** 10.3390/antibiotics13040300

**Published:** 2024-03-26

**Authors:** Attilio Di Maio, Hamza Olleik, Elise Courvoisier-Dezord, Sophie Guillier, Fabienne Neulat-Ripoll, Romain Haudecoeur, Jean-Michel Bolla, Magali Casanova, Jean-François Cavalier, Stéphane Canaan, Valérie Pique, Yolande Charmasson, Elias Baydoun, Akram Hijazi, Josette Perrier, Marc Maresca, Maxime Robin

**Affiliations:** 1Aix Marseille University, University Avignon, CNRS, IRD, IMBE, 13013 Marseille, France; attilio.di-maio@univ-amu.fr (A.D.M.); valerie.pique@univ-amu.fr (V.P.); 2Aix Marseille University, CNRS, Centrale Marseille, iSm2, 13013 Marseille, Franceelise.courvoisier-dezord@univ-amu.fr (E.C.-D.); yolande.charmasson@univ-amu.fr (Y.C.); josette.perrier@univ-amu.fr (J.P.); 3Aix Marseille University, INSERM, SSA, MCT, 13385 Marseille, France; sophie.guillier@intradef.gouv.fr (S.G.); fabienne.ripoll@gmail.com (F.N.-R.); jean-michel.bolla@univ-amu.fr (J.-M.B.); 4University Grenoble Alpes, CNRS, DPM, 38000 Grenoble, France; romain.haudecoeur@univ-grenoble-alpes.fr; 5Aix-Marseille University, CNRS, LISM UMR7255, IMM FR3479, 13009 Marseille, France; magali.casanova@univ-amu.fr (M.C.); jfcavalier@imm.cnrs.fr (J.-F.C.); canaan@imm.cnrs.fr (S.C.); 6Department of Biology, American University of Beirut, Beirut 1107, Lebanon; eliasbay@aub.edu.lb; 7Plateforme de Recherche et D’analyse en Sciences de L’environnement (EDST-PRASE), Beirut 1107, Lebanon; akram.hijazi@ul.edu.lb

**Keywords:** aurone, anti-bacterial agents, cytotoxicity tests, structure-activity relationship

## Abstract

The development of new and effective antimicrobial compounds is urgent due to the emergence of resistant bacteria. Natural plant flavonoids are known to be effective molecules, but their activity and selectivity have to be increased. Based on previous aurone potency, we designed new aurone derivatives bearing acetamido and amino groups at the position 5 of the A ring and managing various monosubstitutions at the B ring. A series of 31 new aurone derivatives were first evaluated for their antimicrobial activity with five derivatives being the most active (compounds **10**, **12**, **15**, **16**, and **20**). The evaluation of their cytotoxicity on human cells and of their therapeutic index (TI) showed that compounds **10** and **20** had the highest TI. Finally, screening against a large panel of pathogens confirmed that compounds **10** and **20** possess large spectrum antimicrobial activity, including on bioweapon BSL3 strains, with MIC values as low as 0.78 µM. These results demonstrate that 5-acetamidoaurones are far more active and safer compared with 5-aminoaurones, and that benzyloxy and isopropyl substitutions at the B ring are the most promising strategy in the exploration of new antimicrobial aurones.

## 1. Introduction

The development of novel antibacterial molecules is a major necessity in the upcoming decades, due to the increasing emergence of multi-drug resistant bacterial strains. This is leading to an elevating mortality rate by infectious disease that may reach more than 10 billion deaths by 2050, according to World Health Organization (WHO) [1]. Among these strains, mycobacteria, in particular *M. tuberculosis*, the pathogenic agent of tuberculosis, are still responsible for 10 million new cases every year worldwide and killed almost 1.5 million patients in 2022. More alarmingly, the number of strains resistant and ultra-resistant to cocktails of antibiotics currently used to treat infection is constantly rising. One of the major concerns is about methicillin resistant *Staphylococcus aureus* (MRSA), but also other Gram-positive and Gram-negative species such as *Acinetobacter baumannii*, *Pseudomonas aeruginosa* or *Klebsiella pneumoniae* [2]. In addition to bacteria, fungi (filamentous fungi such as *Aspergillus fumigatus*, as well as yeasts such as *Candida* species and *Cryptococcus neoformans*) are also responsible for deadly infections, particularly in HIV-infected patients, but also in immunocompetent ones, affecting billions of patients and causing more than 1.5 million deaths per year [3,4,5,6].

Some natural plant molecules and/or their derivatives, including flavonoids, have been reported to possess strong antimicrobial activity. For decades, the biological effect of flavonoids has been studied, focusing on the major subclasses such as flavones, flavonols, flavanones and chalcones. However, in the past ten years, the aurone subclass has been demonstrated to display strong biological effects in diverse fields, such as oncology, dermatology, and infectiology [7,8,9]. The natural occurrences of aurones is limited to a limited number of advanced plant species where they play a variety of key roles as flower pigments, antioxidants and as nectar guides [10]. Although some natural aurones such as cephalocerone [11,12] and hispidol [13] have demonstrated antimicrobial activity, aurones exhibiting natural substitution patterns do not generally lead to the most effective antimicrobial agents. On the other hand, a series of synthetical aurone derivatives showed a strong effect against Gram-positive bacteria [14]. Structurally, aurones are characterized by a benzofuran moiety bearing in position 2 a benzylidene type substituent (rings A, B, and C) (Figure 1). Overall, research on scaffold modifications has mainly focused on the substitution at the B-ring scaffold, e.g., with the introduction of either ferrocene [15], 5-nitroimidazole [16] or quinoline [17] groups, with a global tendency to retain naturally present hydroxy groups at the A-ring.

In the present study, modifications of the A-ring were performed by substitution with an amino group combined with substitutions of the B-ring (Figure 1). The 31 new aurone derivatives obtained (Table 1) were tested in terms of antibacterial and antifungal activities.

## 2. Results

Antimicrobial effect of the aurone derivatives was first evaluated by the determination of their Minimum Inhibitory Concentrations (MICs) on five different bacterial strains representative of Gram-positive bacteria (*Bacillus subtilis* and *Staphylococcus aureus*), Gram-negative bacteria (*Escherichia coli* and *Pseudomonas aeruginosa*), mycobacteria (*Mycobacterium smegmatis*), and one fungal strain (*Candida albicans*) (µM) (Table 2).

In this first screening, among the 31 aurone derivatives tested, compounds **10**, **12**, **15**, **16**, and **20** were the most active (Figure 2). Compound **10** gave the lowest MIC values on all micro-organisms tested in Table 2 (i.e., Gram-positive and -negative bacteria, mycobacteria and fungi), with MICs ranging from 3.12 to 50 µM. Similarly, compound **20** was active on all tested species (MIC ranging from 12.5 to 50 µM), except *P. aeruginosa* for which MIC was superior to 100 µM. Compound **16**, although active on *B. subtilis*, *S. aureus*, *E. coli*, and *M. smegmatis* (MIC ranging from 25 to 50 µM), was however inactive on *P. aeruginosa* and *C. albicans*. Finally, although compounds **12** and **15** were active on tested Gram-positive bacteria and mycobacteria (MIC ranging from 25 to 100 µM and from 50 to 100 µM, respectively), they were inactive on tested Gram-negative bacteria and *C. albicans*. Based on MIC values on this first screening, the observed order of antimicrobial activities is as follows: compound **10** > **20** > **12** = **16** > **15**.

The safety of the five most active derivatives (i.e., compounds **10**, **12**, **15**, **16**, and **20**) was then evaluated using different human cell types (Figure 3 and Figure 4, Table 3).

Overall, compounds **10**, **12**, **15**, **16**, and **20** were found to be safe with most of their CC_50_ values higher than their MIC ones. Compounds **10** and **20** were the safest molecules with mean CC_50_ of 321.5 and 305.1 µM (ranging from 169.0 to 472.4 µM and from 125.9 to >500 µM for compounds **10** and **20**, respectively). Compounds **12**, **15**, and **16** were more toxic, with mean CC_50_ of 129.3, 130.6, and 218.0 µM, respectively. The highest safety of compounds **10** and **20** was further demonstrated when comparing the therapeutic indexes (TI) of the five aurones. Indeed, when calculating the TI of each aurone (by dividing their CC_50_ on human cells (Table 3) by their MIC on *S. aureus* (Table 2)), compounds **10** and **20** gave the highest TI values (ranging from 13.5 to 37.7 and from 10.0 to >40, for compounds **10** and **20**, respectively) compared to the TI values of compounds **12**, **15**, and **16** (Table 4) (ranging from 0.4 to 18.0).

Based on antimicrobial activity and toxicity data, the two most active and safest compounds were identified as compounds **10** and **20**, which were then tested on a larger panel of bacterial and fungal species in order to evaluate their spectrum of activity (Table 5).

The results of this second screening confirmed that compounds **10** and **20** are primarily active on Gram-positive bacteria, with MIC values as low as 0.78 and 3.12 µM for compounds **10** and **20**, respectively. Compounds **10** and **20** were particularly active on the various foodborne pathogens. For example, MIC values of 3.12 and 6.25 µM were obtained on *Listeria monocytogenes* for compounds **10** and **20**, respectively. Compounds **10** and **20** gave good activity against Clostridi such as *C. difficile* (MIC values of 12.5 and 3.12 µM for compounds **10** and **20**, respectively) and *C. botulinum* (MIC values of 0.78 and 3.12 µM for compounds **10** and **20**, respectively). In addition, good activities were also observed on the WHO group 3 pathogen *Bacillus anthracis*, responsible for anthrax disease and used as biological weapon (MIC values of 12.5 and 6.25 µM for compounds **10** and **20**, respectively). These MIC values were consistent with the ones obtained on two other *Bacillus* species, i.e., *B. cereus* and *B. subtilis*. On the other hand, *C. perfringens*, *Enterococcus* species, *P. acnes*, and *S. pyogenes* were found to be weakly sensitive to insensitive to compounds **10** and **20** with MIC values from 50 µM to >100 µM. Overall, compound **10** was more active than compound **20** on most tested Gram-positive bacterial strains, except *C. difficile*, *E. faecium*, and *B. anthracis*, for which compound **20** was more active. *S. aureus* and methicillin-resistant *S. aureus* (MRSA) showed good sensitivity with MIC of 12.5–25 µM, showing that resistance to methicillin did not affect the activity of compounds **10** and **20**.

Compounds **10** and **20** were also active on Gram-negative bacteria, with *A. baumannii*, *E. coli*, and *H. pylori* being the more sensitive strains (MIC values as low as 12.5 and 25 µM for compounds **10** and **20**, respectively)*. S*. *enterica*, *S. flexneri*, and *V. alginolyticus* were also found to be sensitive, with MICs ranging from 25 to 50 µM. Although *P. aeruginosa* was sensitive to compound **10** (MIC of 25 µM), it was insensitive to compound **20** (MIC > 100 µM). Good activities were also obtained on the WHO group 3 pathogens *Brucella melitensis*, *Francisella tularensis*, and *Yersinia pestis* (MIC values as low as 12.5 µM). *E. cloacae* and *K. pneumoniae* were insensitive to compounds **10** and **20** (MIC > 100 µM). Compound **10** was more active than compound **20** on most Gram-negative strains tested, except for *B. melitensis* and *F. tularensis*, for which compound **20** gave lower MIC values.

Although compounds **10** and **20** were active on *M. smegmatis* (Table 2), they were inactive on tested pathogenic mycobacteria, i.e., *M. abscessus* and *M. tuberculosis*.

Finally, in term of antifungal effect, compounds **10** and **20** were active on the filamentous fungi *F. oxysporum* (MIC of 25 µM) but inactive on another important human pathogen *A. fumigatus*. Antifungal activity was also observed on yeasts, including various *Candida species* (*C. albicans*, *C. auris*, *C. glabrata*, and *C. tropicalis*) and *Cryptococcus neoformans* with MIC values as low as 25 and 12.5 µM for compounds **10** and **20**, respectively. In most cases, compounds **10** and **20** gave the same MIC values except for *C. auris*, for which compound **20** was more active than compound **10** (MIC of 12.5 and 50 µM, respectively). *C. tropicalis* had a less sensitive MIC value of 100 µM.

The therapeutic indexes (TI) values of compounds **10** and **20** were calculated using the MIC values reported in Table 5 and the CC_50_ on human cells reported in Table 3 (Table 6 and Table 7).

TI values ranged from 216.6 to 605.6 and from 40.3 to >160.2 for compounds **10** and **20**, respectively, confirming that these two aurone derivatives possess good therapeutic values. Compound **10** was the safest in all cases.

## 3. Discussion

In the present study, 31 new aurone derivatives were synthetized (Figure 5, see Appendix A) and tested in terms of antimicrobial activity against various bacteria and fungi. These new compounds were obtained by the substitution of the aurone scaffold at position 5 by amino and acetamido groups, and through various substitutions at the 2′, 3′ and 4′ positions. The first screening antimicrobial test performed on representative species of bacteria and fungi allowed us to identify compounds **10**, **12**, **15**, **16**, and **20** as the more active aurones. Comparisons between active and inactive structures afforded insightful information to identify the most interesting substitution. Compound **10** can be compared to compound **11**, **12** and **13**. All these compounds are substituted in position 3′ or 4′ by a benzyloxy group. However, only compound **10** and **12** showed an interesting activity. This suggests that benzyloxy substitution in 3′ may be a key element in the activity of the compound. The same methodology could be used to compare compound **20** with other isopropyloxy compounds such as **18**, **19**, **21** and **22**. These four last mentioned compounds showed a weak or no activity against tested micro-organisms. Similarly to compound **10**, this could indicate that the 4′-isopropyl substitution may be a far better alternative to the other position and again that the acetamido group is more effective than the amino group in position 5. From the results, it can be suggested that the acetamido group is important for the antibacterial capacity of the aurones. Thus, compounds **12** and **16** are both 5-aminoaurones; compounds **15** and especially compounds **10** and **20** are 5-acetamidoaurones and showed far better antibacterial activity. Moreover, four of these compounds possess a ring substitution in position 3′ and 4′. Interestingly, benzyloxy-substituted aurones (i.e., **10** and **12**) seem to only be active in positions 3′ when phenyl-substituted aurones (i.e., **15** and **16**) showed activity when substituted both in 3′ and 4′ position. However, considering the activity of compound **10** compared to **12**, **15** and **16**, benzyloxy substitution must be considered more promising than phenyl substitution. The other benzyloxy- and phenyl-substituted aurones (i.e., **11**, **13**, **14** and **17**) showed no activity. Isopropyloxy-substituted aurones also are an interesting option as shown by compound **20**, which is similar in activity to compound **10**. Again only 4′-isopropyl aurones were active; 3′ and 2′ were inactive on the varieties of pathogens tested. Methyl (i.e., **4**–**9**), fluoro (i.e., **24**–**28**), carboxy (i.e., **31**), trifluoromethyl (i.e., **29** and **30**) and hydroxy (i.e., **32** and **33**) substituted aurones showed no activity and thus should not be considered as privileged substitution in the development of antibacterial aurones. Finally, aurone 34 is also inactive and shows that the 5-acetamido substitution is not enough to produce an antibacterial activity to aurones and that a substitution on the B-ring is mandatory. These five aurones were then tested in term of toxicity against various human cell types. The toxicity data demonstrated that compounds **10** and **20** were the safest ones compared to compounds **12**, **15**, and **16**. Again, 5-acetamido aurones seem to be more interesting as they are safer on human cells than 5-aminoaurones. Compounds **10** and **20** were then further tested on a larger panel of pathogens, including Gram-positive and Gram-negative bacteria. In this second screening, these compounds showed an interesting activity as antimicrobials against Gram-positive strains such as *S. aureus*, *methicillin-resistant S. aureus*, *L. monocytogenes*, *B. subtilis* and *C. difficile* and Gram-negative strains such as *E. coli*, *A. baumannii* and *H. pylori.* The two selected compounds shared some structural similarities, such as the 5-acetamido substitution. Out of all the compounds, 3′-benzyloxy and 4′-isopropyloxy were the most promising substitutions. Compared to previously described aurones active on various Gram-positive bacteria but only two Gram-negative strains (i.e., *H. pylori* or *V. alginolyticus*) [14], compounds **10** and **20** were active on a large number of Gram-positive and -negative bacteria as well as fungi. Structurally, the most active compound synthetized by Olleik et al. [14] was a 5,7-dihydroxyaurone substituted in 4′ by a benzyloxy group and in 3′ by a methoxy group. Again, this shows the promising nature of the benzyloxy substitution and that hydroxy aurones, vastly found in nature, are far less active than synthetic aurones such as amino and acetamidoaurones.

## 4. Materials and Methods

### 4.1. Biology

#### 4.1.1. Microorganism Strains and Growth Conditions

Bacterial and fungi strains used in this study, except when mentioned, were obtained from either the American Type Culture Collection (ATCC), the German Leibniz Institute (DSMZ), or the French Pasteur Institute (CIP) and correspond to reference strains. They were maintained on agar plates using appropriate media and culture conditions (in terms of temperature and aerobic/microaerobic/anaerobic condition) as previously described [14,18]. Regarding BSL-3 strains, *Bacillus anthracis*, *Francisella tularensis*, and *Brucella melitensis* strains were maintained on Chocolate agar PolyViteX (Biomerieux) agar at 37 °C, and at 26 °C for *Yersinia pestis* [19,20]. Regarding mycobacteria, *M. smegmatis* mc^2^155 (ATCC700084) was grown in Middlebrook 7H9 complete medium containing 0.05% Tween-80 and 0.2% Glycerol (7H9-TG) and *M. abscessus* (CIP104536T) S and R morphotypes, were cultured in 7H9-TG containing 10% BBL™ Middlebrook OADC Enrichment (7H9-TG^OADC^) at 37 °C under stirring (200 rpm). *M. tuberculosis* mc^2^6230, a derivative of H37Rv which contains a deletion of the RD1 region and *panCD*, resulting in a pan(−) phenotype, was grown in 7H9-TG^OADC^ supplemented with 24 µg/mL D-panthothenate (Sigma-Aldrich, Lyon, France). Cultures were kept at 37 °C without shaking.

#### 4.1.2. Antimicrobial Activity Assay

The antimicrobial activity of aurones on BSL2 bacteria and fungi was evaluated through determination of the Minimum Inhibitory Concentration (MIC) using two-fold serial dilutions in liquid media following the National Committee of Clinical Laboratory Standards (NCCLS, 1997) as previously described [14,18,21,22]. For BSL-3 bacteria, the MIC of aurones was determined following the Clinical and Laboratory Standards Institute (CLSI) recommendations as previously described [23]. For determining the antimycobacterial activity of the different aurones, the microdilution method was used in sterile 96-well flat-bottom Greiner Bio-One polypropylene microplates with lid (Thermo Fisher Scientific) using the resazurin microtiter assay (REMA) as previously described [24,25]. The concentration of aurones leading to 90% inhibition of mycobacteria growth was defined as the MIC. All experiments were performed independently at least three times.

#### 4.1.3. Cytotoxic Assays

The impact of aurones on the viability of human cells were evaluated as previously described [18,22,26]. Human cells used were kidney epithelial cell line A498 (ATCC^®^ HTB-44), normal lung epithelial cells BEAS-2B (ATCC^®^ CRL-9609), intestinal cell line Caco-2 (ATCC^®^ HTB-37), normal epidermal keratinocytes (HaCaT) (from Creative Bioarray, Shirley, NY, USA), liver cell line HepG2 (ATCC^®^ HB-8065), and normal lung fibroblasts IMR-90 (ATCC^®^ CCL186). Cells were cultured in DMEM supplemented with 10% fetal calf serum (FCS), 1% l-glutamine, and 1% antibiotics (all from Invitrogen (Carlsbad, CA, USA). Cells were routinely grown on 25 cm^2^ flasks and maintained in a 5% CO_2_ incubator at 37 °C. For toxicity assays, human cells grown on 25 cm^2^ flasks were detached using trypsin-EDTA solution (from Thermofisher, Waltham, MA, USA), counted using Malassez counting chamber, diluted in appropriate culture media, and seeded into 96-well cell culture plates (Greiner bio-one, Paris, France) at approximately 10^4^ cells per well. The cells were left to grow for 48–72 h at 37 °C in a 5% CO_2_ incubator until confluence. Media from wells was then aspirated and cells were treated with 100 µL of appropriate culture media containing increasing concentrations of tested aurones (from 0 to 400 µM, 1:2 serial dilutions). Volume of DMSO corresponding to 400 µM of aurones was used as negative control and was found not toxic. The plates were then incubated at 37 °C for 48 h. Resazurin-based in vitro toxicity assay kit (from Sigma-Aldrich, Lyon, France) was then used to assess the viability of the cells following manufacturer’s instructions. Briefly, resazurin stock solution was diluted 1:10 in sterile PBS containing calcium and magnesium (PBS^++^, pH 7.4). Plates were aspirated and 100 µL of the diluted solution was added per well. After 2 h incubation at 37 °C, fluorescence intensity was measured using a microplate reader (Biotek, Synergy Mx, Colmar, France) (excitation wavelength of 530 nm/emission wavelength of 590 nm). The fluorescence values were normalized by the controls (DMSO treated cells) and expressed as percentage of cell viability. The CC_50_ values of aurones (i.e., the concentrations causing a reduction of 50% of the cell viability) were calculated using GraphPad^®^ Prism 7 software (San Diego, CA, USA). Experiments were conducted in triplicate (*n* = 3).

### 4.2. Chemistry

^1^H and ^13^C NMR spectra were recorded on a Bruker Avance III nanobay—300 MHz instrument (Bruker, Bremen, Germany, 300 MHz for ^1^H, 75 MHz for ^13^C). Chemical shifts are reported in ppm relative to the solvent in which the spectrum was recorded [^1^H: δ (d_6_-DMSO) = 2.50 ppm, δ (CDCl_3_) = 7.27 ppm; ^13^C: δ (d_6_-DMSO) = 39.52 ppm, δ (CDCl_3_) = 77.16 ppm], full spectra are presented in Appendix A. Combustion analyses were performed at the analysis facilities of Spectropole (https://fr-chimie.univ-amu.fr/spectropole, accessed on 20 December 2023) with a Thermo Finnigan (San Jose, CA, USA) EA 1112 apparatus; all compounds had purity higher than 95%. Microwave-assisted reactions were performed in a CEM Discover microwave reactor with a focused field (CEM Corporation, Matthews, NC, USA). Silica gel F-254 plates (0.25 mm; Merck, Darmstadt, Germany) were used for thin-layer chromatography (TLC), and silica gel 60 (200–400 mesh; Merck) was used for flash chromatography. Unless otherwise stated, reagents were obtained from commercial sources and were used without further purification.

#### 4.2.1. Synthesis Route A

##### Synthesis of N-(4-methoxyphenyl)acetamide (**1a**)

In a solution of 8 g of m-anisidine and 2.5 eq. of NaOH in 50 mL of ethyl acetate, 6.6 g of acetic anhydride (1.5 eq.) was added dropwise. When the addition was completed, the mixture was heated at 80 °C for 5 h. The solution was cooled and filtered. Under pressure, the solvent was removed, the obtained product was dissolved in ethanol and hexane to precipitate, to obtain 5 g of **1a**.

##### Synthesis of 2-Chloro-1-{2-hydroxy-5-[(1-hydroxyethyl)amino]cyclohexyl}ethan-1-one (**2a**)

A total of 5 g of **1a** and 18 g of AlCl_3_ were dissolved in 30 mL of dichloromethane then at 0 °C, 3.5 eq. of chloroacetyl chloride was added dropwise. When the addition was completed, the solution was heated up to 50 °C for 1 h. The mixture was poured on ice and extracted with ethyl acetate. EtOAc was removed under pressure to obtain **2a**.

##### Synthesis of N-(3-oxo-2,3-dihydro-1-benzofuran-5-yl)acetamide (**3a**)

In a flask, 2 g of **2a** and 1.5 eq. of triethylamine were added to 20 mL of acetonitrile; the solution reacted at 25 °C for 12 h. Solvent was removed under pressure. The left-over mixture was dissolved in EtOAc, washed several times with water then extracted. EtOAc was removed under pressure to obtain **3a**.

##### Synthesis of Substituted 5-Acetamidoaurones

In a flask, 1 mmol of **3a** and 1 mmol of the corresponding benzaldehyde were dissolved in 10 mL of choline chloride/urea. Three drops of 50% KOH solution were added. The mixture was heated at 80 °C for 2 h. Water and HCl were added, then the precipitate was filtered and washed several times with ether to obtain acetamido substituted aurones.

##### Synthesis of Substituted 5-Aminoaurone

A total of 10 mmol of acetamido aurones was added to a mixture of EtOH (20 mL) and 0.5 M HCl (5 mL). The solution was refluxed for 2 h. Upon cooling, the solvent was removed under vacuum and the residue obtained was poured onto iced water (100 mL). The resulting solution was neutralized with NH_4_OH 16% until pH = 7. The precipitate formed was collected by filtration and washed with excess cold water.

#### 4.2.2. Synthesis Route B

##### Synthesis of 4-Acetamidophenyl Acetate (**1b**)

To a solution of 8 g of 4-aminophenol and 2.5 eq. of NaOH in 50 mL of ethyl acetate, 26.5 g of acetic anhydride (3.5 eq.) was added dropwise. When the addition was completed, the solution was heated at 80 °C for 5 h. Upon cooling, the mixture was filtered. Solvent was removed under pressure; the product was recrystallized in ethanol and hexane to obtain 5 g of **1b**.

##### Synthesis of N-(3-acetyl-4-hydroxyphenyl)acetamide (**2b**)

To a solution of **1b** (5 g, 26 mmol), 15 g of AlCl_3_ (113 mmol, 4 eq.) and 1.9 g of KCl (26 mmol, 1 eq.) were added. The mixture was then heated a 165 °C for 1 h until a brown paste appeared. Upon cooling, ice cold water was added (300 mL) and the mixture was filtered to obtain 2 g of a beige powder (**2b**).

##### Synthesis of Substituted of 5-Acetamidochalcones (**3b**)

In a flask, 193 mg of **2b** (0.001 mmol), 1 eq. of chosen benzaldehyde and 188 mg of LiOH (16 eq., 0.016 mmol) were dissolved in ethanol (20 mL). The mixture was heated for 2 h at 90 °C. The solvent was then removed under pressure, cold water and HCl were added, and the precipitate was filtered to obtain the desired chalcone (**3b**).

##### Synthesis of Substituted 5-Acetamidoaurones

To a mixture of chosen **3b** chalcone in pyridine (20 mL), 1 eq. of mercury acetate was added. The solution was heated for 1 h at 110 °C. Water and HCl were added. The precipitate was filtered and washed several times with ice cold water to get rid of the mercury. Obtention of a powder red-yellow powder depended the substitution.

##### Synthesis of Substituted 5-Aminoaurone

For synthesis of 5-aminoaurone see Section Synthesis of Substituted 5-Aminoaurone.

Yield: 83%; mp: 234.8 °C; ^1^H NMR (300 MHz, DMSO-d_6_): δ 10.15 (s, 1H, NH), 8.20–8.18 (dd, 1H, *J* = 1.2;7.8 Hz, C-H_6′_), 8.10 (d, 1H, *J* = 1.9 Hz, C-H_4_), 7.83–7.80 (dd, 1H, *J* = 2.2;8.9 Hz, C-H_6_), 7.52–7.49 (d, 1H, *J* = 8.9 Hz, C-H_7_), 7.46 (dt, 1H, *J* = 7.1 Hz, C-H_4′_), 7.19 (s, 1H, C-H_10_), 7.16–7.09 (m, 2H, C-H_3′,5′_), 3.91 (s, 3H, OCH_3_), 2.07 (s, 3H, NHCOCH_3_). ^13^C NMR (75 MHz, DMSO-d_6_): δ 183.53(C-3), 168.41(CO), 161.21(C-8), 158.32(C-2′), 146.75(C-2) 135.53(C-5), 131.99(C-4′), 131.11(C-6′), 128.71(C-6), 120.89(C-1′), 120.68(C-5′), 120.08(C-9), 113.31(C-3′), 113.15(C-7), 111.57(C-10), 105.47(C-4), 55.84(OCH_3_), 23.83(CH_3_). Elemental analysis calcd (%) for C_18_H_15_NO_4_: C, 69.89; H, 4.89; N, 4.53; found C, 69.87; H, 4.91; N, 4.52. *m*/*z*: 309.1001 (100.0%).


**(Z)-N-(2-(3-methoxybenzylidene)-3-oxo-2,3-dihydrobenzofuran-5-yl)acetamide (5):**


Yield: 71%; mp: 204.2 °C; ^1^H NMR (300 MHz, DMSO-d_6_): δ 10.16 (s, 1H, NH), 8.10 (d, 1H, *J* = 2 Hz, C-H_4_), 7.83–7.80 (dd, 1H, *J* = 2.2;8.9 Hz, C-H_6_), 7.60–7.55 (m, 2H, C-H_2′,4′_), 7.54–7.51 (d, 1H, *J* = 8.9 Hz, C-H_7_), 7.43 (dt, 1H, *J* = 8.0 Hz, C-H_5′_), 7.06–7.03 (dd, 1H, *J* = 2.6;8.2 Hz, C-H_6′_), 6.90 (s, 1H, C-H_10_), 7.16–7.09 (m, 2H, C-H_3′,5′_), 3.82 (s, 3H, OCH_3_), 2.07 (s, 3H, NHCOCH_3_). ^13^C NMR (75 MHz, DMSO-d_6_): δ 183.68(C-3), 168.42(CO), 161.33(C-8), 159.42(C-3′), 146.88(C-2), 135.58(C-5), 133.08(C-6′), 130(C-2′), 128.82(C-6), 123.74(C-1′), 120.59(C-9), 116.57(C-6′), 115.76(C-4′), 113.35(C-7), 113.15(C-4), 112.07(C-10), 55.15(OCH3), 23.83(CH_3_). Elemental analysis calcd (%) for C_18_H_15_NO_4_: C, 69.89; H, 4.89; N, 4.53; found C, 69.84; H, 4.88; N, 4.53. *m*/*z*: 309.1001 (100.0%).


**(Z)-N-(2-(4-methoxybenzylidene)-3-oxo-2,3-dihydrobenzofuran-5-yl)acetamide (6):**


Yield: 91%; mp: 252 °C [1]; ^1^H NMR (300 MHz, DMSO-d_6_): δ 10.17 (s, 1H, NH), 8.10 (d, 1H, *J* = 2 Hz, C-H_4_), 7.96–7.93 (d, 2H, *J* = 7.9 Hz, C-H_2′,6′_), 7.81–7.78 (dd, 1H, *J* = 2.2;8.9 Hz, C-H_6_), 7.50–7.47 (d, 1H, *J* = 8.9 Hz, C-H_7_), 7.08–7.06 (d, 2H, *J* = 8.0 Hz, C-H_3′,5′_), 6.91 (s, 1H, C-H_10_), 3.83 (s, 3H, OCH_3_), 2.07 (s, 3H, NHCOCH_3_). ^13^C NMR (75 MHz, DMSO-d_6_): δ 183.37(C-3), 168.46(CO), 161.06(C-8), 161.00(C-4′), 145.75(C-2), 135.47(C-5), 133.40(C-2′,6′), 128.51(C-6), 124.47(C-1′), 120.98(C-9), 114.71(C-3′-5′), 113.30(C-7), 113.07(C-4), 112.74(C-10), 55.40(OCH3), 23.91(CH3). Elemental analysis calcd (%) for C_18_H_15_NO_4_: C, 69.89; H, 4.89; N, 4.53; found C, 69.78; H, 4.87; N, 4.48. *m*/*z*: 309.1001 (100.0%).


**(Z)-5-amino-2-(2-methoxybenzylidene)benzofuran-3(2H)-one (7):**


Yield: 22%; mp: 189.3 °C; ^1^H NMR (300 MHz, DMSO-d_6_): δ 8.18–8.16 (dd, 1H, *J* = 1.6;7.8 Hz, C-H_6′_), 7.43 (dt, 1H, *J* = 1.5;8.5 Hz, C-H_4′_), 7.26–7.23 (d, 1H, *J* = 8.8 Hz, C-H_7_), 7.14–7.11 (m, 2H, C-H_3′,5′_), 7.10 (s, 1H, C-H_10_), 7.06–7.03 (dd, 1H, *J* = 2.5;8.8 Hz, C-H_6_), 6.84 (d, 1H, *J* = 2.4 Hz, C-H_4_), 5.23 (bs, 2H, NH_2_), 3.90 (s, 3H, OCH_3_). ^13^C NMR (75 MHz, DMSO-d_6_): δ 184.09(C-3), 158.13(C-2′), 158.00(C-8), 147.05(C-2), 145.56(C-5), 131.55(C-4′), 130.98(C-6′), 124.55(C-6), 121.01(C-9, 120.84(C-5′), 120.41(C-1′), 113.14(C-7), 111.47(C-10), 105.44(C-3′), 104.18(C-4), 55.79(OCH_3_). Elemental analysis calcd (%) for C_16_H_13_NO_3_: C, 71.90; H, 4.90; N, 5.24; found C, 71.85; H, 4.95; N, 5.21. *m*/*z*: 267.0895 (100.0%).


**(Z)-5-amino-2-(3-methoxybenzylidene)benzofuran-3(2H)-one (8):**


Yield: 50%; mp: 190 °C; ^1^H NMR (300 MHz, DMSO-d_6_): δ 7.59–7.57 (d, 1H, *J* = 7.7 Hz, C-H_6′_), 7.54 (d, 1H, *J* = 2.4 Hz, C-H_4_), 7.49–7.46 (d, 1H, *J* = 8.8 Hz, C-H_7_), 7.42 (dt, 1H, *J* = 8.2 Hz, C-H_5′_), 7.38–7.35 (dd, 1H, *J* = 2.2;8.8 Hz, C-H_6_), 7.24 (d, 1H, *J* = 2.11 Hz, C-H_2′_), 7.06–7.03 (dd, 1H, *J* = 1.9;8.1 Hz, C-H_4′_), 6.88 (s, 1H, C-H_10_), 3.82 (s, 3H, OCH_3_). ^13^C NMR (75 MHz, DMSO-d_6_): δ 183.6 (C-3), 160.79 (C-8), 159.42 (C-3′), 146.96 (C-2), 137.91 (C-5′), 133.13 (C-1′), 129.99 (C-5′), 127.97(C-6), 123.71 (C-7), 121.18 (C-9), 116.54 (C-6′), 115.72 (C-4′), 113.87 (C-4), 111.88 (C-2′), 111.18 (C-10), 55.16 (OCH_3_). Elemental analysis calcd (%) for C_16_H_13_NO_3_: C, 71.90; H, 4.90; N, 5.24; found C, 71.92; H, 4.92; N, 5.28. *m*/*z*: 267.0895 (100.0%).


**(Z)-5-amino-2-(4-methoxybenzylidene)benzofuran-3(2H)-one (9):**


Yield: 86%; mp: 110.4 °C; ^1^H NMR (300 MHz, DMSO-d_6_): δ 7.95–7.92 (d, 2H, *J* = 8.1 Hz, C-H_2′_), 7.31–7.28 (d, 1H, *J* = 8.8 Hz, C-H_7_), 7.13–7.10 (dd, 1H, *J* = 2.2;8.8 Hz, C-H_6_), 7.09–7.06 (d, 2H, *J* = 8.1 Hz, C-H_3′_), 6.93 (d, 1H, *J* = 2.4 Hz, C-H_4_), 6.83 (s, 1H, C-H_10_), 3.82 (s, 3H, OCH_3_). ^13^C NMR (75 MHz, DMSO-d_6_): δ 183.77 (C-4), 160.61 (C-4′), 158.51 (C-8), 145.94 (C-5), 143.60 (C-2), 133.16 (C-2′), 125.22 (C-1′), 124.69 (C-6), 121.35 (C-9), 114.65 (C-3′), 113.30 (C-4), 111.68 (C-10), 106.77 (C-7), 55.37 (OCH_3_). Elemental analysis calcd (%) for C_16_H_13_NO_3_: C, 71.90; H, 4.90; N, 5.24; found C, 71.88; H, 4.97; N, 5.30. *m*/*z*: 267.0895 (100.0%).


**(Z)-N-(2-(3-(benzyloxy)benzylidene)-3-oxo-2,3-dihydrobenzofuran-5-yl)acetamide (10):**


Yield: 80%; mp: 204.5 °C; ^1^H NMR (300 MHz, DMSO-d_6_): δ 10.16 (s, 1H, NH), 8.10 (d, 1H, *J* = 2 Hz, C-H_4_), 7.84–7.80 (dd, 1H, *J* = 2.2;8.9 Hz, C-H_6_), 7.63 (bs, 1H, C-H_2′_), 7.58–7.32 (m, 8H,), 7.14–7.11 (dd, 1H, *J* = 7.9 Hz, C-H_4′_), 6.89 (s, 1H, C-H_10_), 5.18 (m, 2H, CH_2_), 2.07 (s, 3H, NHCOCH_3_). ^13^C NMR (75 MHz, DMSO-d_6_): δ 183.74 (C-4), 168.51 (CO), 161.36 (C-8), 158.53 (C-3′), 146.92 (C-2), 136.87 (C-1bn), 135.64 (C-5), 133.14 (C-5′), 130.1 (C-6′), 128.87 (C-6), 128.48 (C-3bn), 127.94 (C-4bn), 127.83 (C-2bn), 124.14 (C-1′), 120.62 (C-9), 117.27 (C-4′), 116.75 (C-2′), 113.45 (C-7), 113.17 (C-4), 112.09 (C-10), 69.34 (CH2), 23.91 (CH3). Elemental analysis calcd (%) for C_24_H_19_NO_4_: C, 74.79; H, 4.97; N, 3.63; found C, 74.74; H, 5.01; N, 3.60. *m*/*z*: 385.13141 (100.0%).


**(Z)-N-(2-(4-(benzyloxy)benzylidene)-3-oxo-2,3-dihydrobenzofuran-5-yl)acetamide (11):**


Yield: 72%; mp: 212.8 °C; ^1^H NMR (300 MHz, DMSO-d_6_): δ 10.14 (s, 1H, NH), 8.09 (d, 1H, *J* = 2.02 Hz, C-H_4_), 7.96–7.93 (d, 2H, *J* = 8.8 Hz, C-H_2′_), 7.82–7.79 (dd, 1H, *J* = 2.2;8.9 Hz, C-H_6_), 7.50–7.34 (m, 6H, C-H_7, bn_), 7.16–7.13 (d, 2H, *J* = 8.8 Hz, C-H_3′_), 6.90 (s, 1H, C-H_10_), 5.18 (m, 2H, CH_2_), 2.07 (s, 3H, NHCOCH_3_).

^13^C NMR (75 MHz, DMSO-d_6_): δ 183.37 (C-3), 168.48 (CO), 161.06 (C-8), 159.93 (C-4′), 145.75 (C-2), 136.58 (C-1bn), 135.45 (C-5), 133.39 (C-2′), 128.56 (C-1′), 128.49 (C-3bn), 128 (C-4bn), 127.84 (C-2bn), 124.64 (C-6), 120.95 (C-9), 115.52 (C-3′), 113.33 (C-7), 113.08 (C-4), 112.68 (C-10), 69.44 (CH2), 23.9 (CH3). Elemental analysis calcd (%) for C_24_H_19_NO_4_: C, 74.79; H, 4.97; N, 3.63; found C, 74.77; H, 4.96; N, 3.61. *m*/*z*: 385.13141 (100.0%).


**(Z)-5-amino-2-(3-(benzyloxy)benzylidene)benzofuran-3(2H)-one (12):**


Yield: 80%; mp: 132.6 °C; ^1^H NMR (300 MHz, DMSO-d_6_): δ 7.61 (bs, 1H, C-H_2′_), 7.55–7.53 (d, 1H, *J* = 8.8 Hz, C-H_7_), 7.50–7.48 (d, 2H, C-H_2bn_), 7.41 (dt, 2H, C-H_3bn_), 7.36–7.33 (m, 2H, C-H_5′,4bn_), 7.11–7.09 (dd, 1H, *J* = 7.9 Hz, C-H_6_), 7.06–7.04 (dd, 1H, *J* = 7.9 Hz, C-H_4′_), 6.83 (d, 1H, *J* = 2 Hz, C-H_4_), 6.77 (s, 1H, C-H_10_), 5.26 (bs, 2H, NH_2_), 5.17 (m, 2H, CH_2_). ^13^C NMR (75 MHz, DMSO-d_6_): δ 184.24 (C-3), 158.48 (C-3′), 158.14 (C-8), 147.21 (C-2), 145.63 (C-5), 136.87 (C-1bn), 133.43 (C-1′), 129.97 (C-5′), 128.42 (C-3bn), 127.86 (C-4bn), 127.75 (C-2bn), 124.71 (C-6), 123.89 (C-7), 120.91 (C-9), 117.04 (C-4′), 116.4 (C-2′), 113.21 (C-6′), 110.73 (C-10), 105.45 (C-4), 69.31 (CH2). Elemental analysis calcd (%) for C_22_H_17_NO_3_: C, 76.95; H, 4.99; N, 4.08; found C, 76.88; H, 5.01; N, 4.04. *m*/*z*: 343.12084 (100.0%).


**(Z)-5-amino-2-(4-(benzyloxy)benzylidene)benzofuran-3(2H)-one (13):**


Yield: 80%; mp: 141.6 °C; ^1^H NMR (300 MHz, DMSO-d_6_): δ 7.97–7.95 (d, 2H, C-H_2′-6′_
*J* = 8.8 Hz), 7.49–7.47 (m, 3H, C-H), 7.41–7.39 (m, 4H, C-H_bn_), 7.28 (s, 1H, C-H_4_), 7.17–7.15 (d, 2H, C-H_3′-5_, *J* = 8.8 Hz), 6.92 (s, 1H, C-H_10_), 5.20 (s, 2H, CH_2_). ^13^C NMR (75 MHz, DMSO-d_6_): δ 183.16 (C-3), 160.82 (C-4′), 159.91 (C-8), 145.76 (C-2), 136, 55 (C-5), 133.36 (C-1′), 128.46 (C-3bn, C-5bn), 128.20 (C-1bn), 127.92 (C-4bn), 127.81 (C-2bn, C-6bn), 124.64 (C-6), 121.55 (C-9), 115.5 (C-2′, C-6′), 113.90 (C-7),112.58 (C-4), 111.79 (C-10), 69.42 (CH_2_). (CH3). Elemental analysis calcd (%) for C_22_H_17_NO_3_: C, 76.95; H, 4.99; N, 4.08; found C, 76.77; H, 4.98; N, 4.06. *m*/*z*: 343.13141 (100.0%).


**(Z)-N-(3-oxo-2-(3-phenoxybenzylidene)-2,3-dihydrobenzofuran-5-yl)acetamide (14):**


Yield: 96%; mp: 196.4 °C; ^1^H NMR (300 MHz, DMSO-d_6_): δ 10.16 (s, 1H, NH), 8.10 (d, 1H, *J* = 2 Hz, C-H_4_), 7.83–7.79 (dd, 1H, *J* = 2.2;8.9 Hz, C-H_6_), 7.75–7.72 (d, 1H, *J* = 7.9 Hz, C-H_6′_), 7.67 (bs, 1H, C-H_2′_), 7.51 (dt, 1H, *J* = 7.4 Hz, C-H_5′_), 7.44–7.41 (m, 3H, C-H_4′,8′_), 7.20 (t, 1H, *J* = 7.4 Hz, C-H_10′_), 7.10–7.07 (d, 3H, C-H_9′, 7_), 6.93 (s, 1H, C-H_10_), 2.07 (s, 3H, NHCOCH_3_). ^13^C NMR (75 MHz, DMSO-d_6_): δ 183.72 (C-3), 168.52 (CO), 161.30 (C-8), 157.14 (C-7′), 156.19 (C-3′), 147.11 (C-2), 135.68 (C-5), 133.78 (C-1′), 130.60 (C-6), 130.17 (C-9′), 128.92 (C-5′), 126.54 (C-9), 123.88 (C-4), 120.62 (C-6′), 120.57 (C-10′), 119.97 (C-4′), 118.99 (C-8′), 113.30 (C-2′), 113.21 (C-7), 111.41 (C-10), 23.91 (CH3). Elemental analysis calcd (%) for C_23_H_17_NO_4_: C, 74.38; H, 4.61; N, 3.77; found C, 74.33; H, 4.63; N, 3.71. *m*/*z*: 371.11576 (100.0%).


**(Z)-N-(3-oxo-2-(4-phenoxybenzylidene)-2,3-dihydrobenzofuran-5-yl)acetamide (15):**


Yield: 67%; mp: 213.5 °C; ^1^H NMR (300 MHz, DMSO-d_6_): δ 10.15 (s, 1H, NH), 8.11 (d, 1H, *J* = 2 Hz, C-H_4_), 8.03–8.00 (d, 2H, *J* = 7.9 Hz, C-H_2′_), 7.83–7.79 (dd, 1H, *J* = 2.2;8.9 Hz, C-H_6_), 7.50–7.42 (m, 3H, C-H_7,7′_), 7.22 (t, 1H, *J* = 7.4 Hz, C-H_8′_), 7.12–7.09 (m, 4H, C-H_3′,6′_), 6.94 (s, 1H, C-H_10_), 2.07 (s, 3H, NHCOCH_3_). ^13^C NMR (75 MHz, DMSO-d_6_): δ 183.43 (C-3), 168.38 (CO), 161.14 (C-8), 158.41 (C-1″), 155.43 (C-4′), 146.18 (C-2), 135.5 (C-5), 133.43 (C-3″), 130.15 (C-2′), 128.67 (C-6), 126.79 (C-1′), 124.27 (C-4″), 120.76 (C-9), 119.41 (C-3′), 118.28 (C-2″), 113.24 (C-7), 113.11 (C-4), 111.83 (C-10), 23.82 (CH3). Elemental analysis calcd (%) for C_23_H_17_NO_4_: C, 74.38; H, 4.61; N, 3.77; found C, 74.35; H, 4.67; N, 3.73. *m*/*z*: 371.11576 (100.0%).


**(Z)-5-amino-2-(3-phenoxybenzylidene)benzofuran-3(2H)-one (16):**


Yield: 51%; mp: 145.6 °C; ^1^H NMR (300 MHz, DMSO-d_6_): δ 7.71–7.69 (d, 1H, *J* = 7.9 Hz, C-H_6′_), 7.64 (bs, 1H, C-H_2′_), 7.51–7.49 (d, 1H, *J* = 7.9 Hz, C-H_6_), 7.44 (dt, 2H, C-H_9′_), 7.22–7.15 (m, 2H, C-H_4,5′_), 7.09–7.04 (m, 4H, C-H_7,9′,10′_), 6.83 (d, 1H, *J* = 2.02 Hz, C-H_4_), 6.77 (s, 1H, C-H_10_), 5.41 (bs, 2H, NH_2_). ^13^C NMR (75 MHz, DMSO-d_6_): δ 184.24(C-3), 158.13(C-7′), 157.1(C-3′), 156.21(C-8), 147.41(C-2), 145.56(C-5), 134.09(C-1′), 130.52(C-5′), 130.15(C-9′), 126.34(C-6), 124.82(C-9), 123.81(C-6′), 120.88(C-10′), 120.43(C-4′), 119.65(C-4), 118.95(C-8′), 113.13(C-8), 110.13(C-10), 105.61(C-7). Elemental analysis calcd (%) for C_21_H_15_NO_3_: C, 76.58; H, 4.59; N, 4.25; found C, 76.56; H, 4.61; N, 4.22. *m*/*z*: 329.10519 (100.0%).


**(Z)-5-amino-2-(4-phenoxybenzylidene)benzofuran-3(2H)-one (17):**


Yield: 76%; mp: 170.6 °C; ^1^H NMR (300 MHz, DMSO-d_6_): δ 7.99–7.96 (d, 2H, *J* = 8.6 Hz, C-H_3′_), 7.44 (dt, 2H, *J* = 7.7 Hz, C-H_7′_), 7.24–7.21 (m, 2H, C-H_7,8′_), 7.10–7.02 (m, 5H, C-H_6,2′,6′_), 6.83 (bs, 2H, C-H_4,10)_, 5.25 (bs, 2H, NH_2_). ^13^C NMR (75 MHz, DMSO-d_6_): δ 184.11 (C-3), 158.16 (C-5′), 158.02 (C-8), 155.58 (C-4′), 146.56 (C-2), 145.6 (C-5), 133.27 (C-7′), 130.24 (C-2′), 127.21 (C-1′), 124.62 (C-6), 124.28 (C-8′), 121.15 (C-9), 119.41 (C-3′), 118.39 (C-6′), 113.17 (C-4), 110.62 (C-10), 105.45 (C-7). Elemental analysis calcd (%) for C_21_H_15_NO_3_: C, 76.58; H, 4.59; N, 4.25; found C, 76.43; H, 4.64; N, 4.54. *m*/*z*: 329.10519 (100.0%).


**(Z)-N-(2-(2-isopropoxybenzylidene)-3-oxo-2,3-dihydrobenzofuran-5-yl)acetamide (18):**


Yield: 93%; mp: 230 °C; ^1^H NMR (300 MHz, DMSO-d_6_): δ 10.15 (s, 1H, NH), 8.21–8.19 (d, 1H, *J* = 1.2;7.8 Hz, C-H_6′_), 8.12 (bs, 1H, C-H_4_), 7.81–7.78 (d, 1H, *J* = 8.7 Hz, C-H_6_), 7.52–7.49 (d, 1H, *J* = 8.9 Hz, C-H_7_), 7.42 (dt, 1H, *J* = 7.7 Hz, C-H_4′_), 7.20 (s, 1H, C-H_10_), 7.16–7.13 (d, 1H, *J* = 8.3 Hz, C-H_3′_), 7.08 (dt, 1H, *J* = 7.6 Hz, C-H_5′_), 4.75 (q, 1H, *J* = 5.9;11.9 Hz, C-H_isop_), 2.07 (s, 3H, NHCOCH_3_), 1.35–1.33 (d, 6H, *J* = 5.8 Hz, C-H3_isop_). ^13^C NMR (75 MHz, DMSO-d_6_): δ 183.52 (C-3), 168.39 (CO), 161.15 (C-8), 156.76 (C-2′), 146.69 (C-2), 135.5 (C-5), 131.85 (C-6′), 131.38 (C-4′), 128.64 (C-6), 121.03 (C-9), 120.73 (C-1′), 120.69 (C-5′), 113.85 (C-7), 113.28 (C-4), 113.11 (C-10), 105.92 (C-3′), 70.49 (CHiPr), 23.83 (CH3), 21.74 (CH3iPr). Elemental analysis calcd (%) for C_20_H_19_NO_4_: C, 71.20; H, 5.68; N, 4.15; found C, 71.14; H, 5.67; N, 4.12. *m*/*z*: 337.13 (100.0%).


**(Z)-N-(2-(3-isopropoxybenzylidene)-3-oxo-2,3-dihydrobenzofuran-5-yl)acetamide (19):**


Yield: 91%; mp: 167 °C; ^1^H NMR (300 MHz, DMSO-d_6_): δ 10.19 (s, 1H, NH), 8.12 (d, 1H, *J* = 2 Hz, C-H_4_), 7.83–7.80 (dd, 1H, *J* = 2.2; 8.9 Hz, C-H_6_), 7.57–7.52 (m, 3H, C-H_2′,4′,7_), 7.40 (dt, 1H, *J* = 8.0 Hz, C-H_5′_), 7.04–7.01 (dd, 1H, *J* = 2.6;8.2 Hz, C-H_6′_), 6.91 (s, 1H, C-H_10_), 4.68 (q, 1H, *J* = 5.9;11.9 Hz, C-H_isop_), 2.07 (s, 3H, NHCOCH_3_), 1.31–1.29 (d, 6H, *J* = 5.8 Hz, C-H3_isop_). ^13^C NMR (75 MHz, DMSO-d_6_): δ 183.73 (C-3), 168.49 (CO), 161.35 (C-8), 157.66 (C-3′), 146.87 (C-2), 135.62 (C-5), 133.17 (C-1′), 130.1 (C-5′), 128.86 (C-6), 123.56 (C-6′), 120.64 (C-9), 118.25 (C-2′), 117.32 (C-3′), 113.41 (C-7), 113.17 (C-4), 112.28 (C-10), 69.35 (CHiPr), 23.88 (CH3), 21.77 (CH3iPr). Elemental analysis calcd (%) for C_20_H_19_NO_4_: C, 71.20; H, 5.68; N, 4.15; found C, 71.18; H, 5.66; N, 4.16. *m*/*z*: 337.13 (100.0%).


**(Z)-N-(2-(4-isopropoxybenzylidene)-3-oxo-2,3-dihydrobenzofuran-5-yl)acetamide (20):**


Yield: 93%; mp: 200.2 °C; ^1^H NMR (300 MHz, DMSO-d_6_): δ 10.14 (s, 1H, NH), 8.10 (d, 1H, *J* = 1.9 Hz, C-H_4_), 7.94–7.91 (d, 2H, *J* = 8.8 Hz, C-H_2′_), 7.82–7.78 (dd, 1H, *J* = 2.2;8.8 Hz, C-H_6_), 7.51–7.48 (d, 1H, *J* = 8.89 Hz, C-H_7_), 7.06–7.03 (d, 2H, *J* = 8.8 Hz, C-H_3′_), 6.90 (s, 1H, C-H_10_), 4.72 (q, 1H, *J* = 5.9;11.9 Hz, C-H_isop_), 2.07 (s, 3H, NHCOCH_3_), 1.30–1.28 (d, 6H, *J* = 5.8 Hz, C-H3_isop_). ^13^C NMR (75 MHz, DMSO-d_6_): δ 183.25 (C-3), 168.39 (CO), 160.98 (C-8), 159.16 (C-4′), 145.6 (C-2), 135.39 (C-5), 133.41 (C-2′), 128.48 (C-6), 124.02 (C-1′), 120.94 (C-9), 115.94 (C-3′), 113.2 (C-7), 113.06 (C-4), 112.77 (C-10), 69.5 (CHiPr), 23.83 (CH3), 21.68 (CH3iPr). Elemental analysis calcd (%) for C_20_H_19_NO_4_: C, 71.20; H, 5.68; N, 4.15; found C, 71.18; H, 5.69; N, 4.12. *m*/*z*: 337.13 (100.0%).


**(Z)-5-amino-2-(2-isopropoxybenzylidene)benzofuran-3(2H)-one (21):**


Yield: 68%; mp: 126.7 °C; ^1^H NMR (300 MHz, DMSO-d_6_): δ 8.19–8.17 (d, 1H, *J* = 6.9 Hz, C-H_6′_), 7.39 (t, 1H, *J* = 7.3 Hz, C-H_4′_), 7.26–7.24 (d, 1H, *J* = 8.7 Hz, C-H_7_), 7.14 (d, 1H, C-H_3′_), 7.11 (bs, 2H, NH2), 7.08–7.06 (d, 1H, C-H_5′_), 7.06–7.04 (dd, 1H, *J* = 2.2;7.7 Hz, C-H_6_), 6.95 (s, 1H, C-H_10_), 6.85 (d, 1H, *J* = 2.2 Hz, C-H_4_), 4.73 (q, 1H, *J* = 5.9;11.9 Hz, C-H_isop_), 1.34–1.32 (d, 6H, *J* = 5.8 Hz, C-H3_isop_). ^13^C NMR (75 MHz, DMSO-d_6_): 184.16 (CO), 158.14 (C-8), 156.62 (C-2′), 147.02 (C-2), 145.24 (C-5), 131.55 (C-6′), 131.33 (C-4′), 124.76 (C-6), 121.4 (C-9), 121.11 (C-5′), 120.72 (C-1′), 113.85 (C-7), 113.23 (C-10), 105.77 (C-3′), 104.8 (C-4), 70.46 (CHiPr), 21.82 (CH3iPr). Elemental analysis calcd (%) for C_20_H_19_NO_4_: C, 71.20; H, 5.68; N, 4.15; found C, 71.24; H, 5.74; N, 4.18. *m*/*z*: 337.13 (100.0%).


**(Z)-5-amino-2-(3-isopropoxybenzylidene)benzofuran-3(2H)-one (22):**


Yield: 51%; mp: 198.4 °C; ^1^H NMR (300 MHz, DMSO-d_6_): δ 7.57–7.54 (d, 1H, *J* = 7.8 Hz, C-H_6′_), 7.51 (bs, 1H, C-H_2′_), 7.51–7.48 (d, 1H, *J* = 8.6 Hz, C-H_7_), 7.43–7.40 (d, 2H, *J* = 7.8 Hz, C-H_4′_), 7.41 (dt, 1H, *J* = 7.8 Hz, C-H_5′_), 7.27 (d, 1H, *J* = 1.9 Hz, C-H_4_), 7.04–7.01 (dd, 1H, *J* = 2.0;8.0 Hz, C-H_6_), 6.89 (s, 1H, C-H_10_), 4.68 (q, 1H, *J* = 5.9;11.9 Hz, C-H_isop_), 1.31–1.29 (d, 6H, *J* = 5.8 Hz, C-H3_isop_). ^13^C NMR (75 MHz, DMSO-d_6_): δ 183.65 (C-3), 160.93 (C-8), 157.67 (C-3′), 146.95 (C-2), 137.64 (C-5), 133.24 (C-1′), 130.11 (C-5′), 128.17 (C-6), 123.56 (C-6′), 121.25 (C-9), 118.24 (C-2′), 117.28 (C-3′), 113.96 (C-7), 112.14 (C-4), 111.46 (C-10), 69.35 (CHiPr), 21.79 (CH3iPr). Elemental analysis calcd (%) for C_10_H_17_NO_3_: C, 71.20; H, 5.68; N, 4.15; found C, 71.18; H, 5.65; N, 4.12. *m*/*z*: 295.12 (100.0%).


**(Z)-5-amino-2-(4-isopropoxybenzylidene)benzofuran-3(2H)-one (23):**


Yield: 50%; mp: >350 °C; ^1^H NMR (300 MHz, DMSO-d_6_): δ 7.97–7.95 (d, 2H, *J* = 8.8 Hz, C-H_2′_), 7.66–7.64 (m, 3H, C-H_4,6,7_), 7.07–7.05 (d, 2H, *J* = 8.8 Hz, C-H_3′_), 6.98 (s, 1H, C-H_10_). ^13^C NMR (75 MHz, DMSO-d_6_): δ 183.22 (C-3), 160.22 (C-8), 159.12 (C-4′), 145.7 (C-2), 133.38 (C-2′), 138.33 (C-5), 127.35 (C-7), 124.1 (C-1′), 121.52 (C-9), 115.94 (C-3′), 113.69 (C-6), 112.51 (C-4), 110.58 (C-10), 69.49 (CH2iPr), 21.68 (CH3iPr). Elemental analysis calcd (%) for C_10_H_17_NO_3_: C, 71.20; H, 5.68; N, 4.15; found C, 71.15; H, 5.66; N, 4.18. *m*/*z*: 295.12 (100.0%).


**(Z)-N-(2-(2-fluorobenzylidene)-3-oxo-2,3-dihydrobenzofuran-5-yl)acetamide (24):**


Yield: 68%; mp: 226 °C; ^1^H NMR (300 MHz, DMSO-d_6_): δ 10.17 (s, 1h, NH), 8.25 (t, 1H, *J* = 7.8 Hz, C-H_2′_), 8.12 (d, 1H, *J* = 2.2 Hz, C-H_4_), 7.85–7.82 (dd, 1H, *J* = 2.5, 8.7 Hz, C-H_6′_), 7.54–7.51 (m, 2H, C-H_4′,7_), 7.41–7.39 (d, 1H, C-H_3′_), 7.35 (dt, 1H, C-H_5′_), 6.90 (s, 1H, C-H_10_), 2.08 (s, 3H, CH3). ^13^C NMR (75 MHz, DMSO-d_6_): δ 183.5 (C-3), 168.44 (CO), 164.9–161.44 (C-2′, *J* = 260 Hz), 161.41 (C-8), 147.71 (C-2), 135.78 (C-5), 132.24–132.12 (C-4′, *J* = 8.8 Hz), 131.31 (C-6′), 129.02 (C-6), 125.11 (C-5′, *J* = 3.3 Hz), 120.38 (C-9), 119.67–119.52 (C-1′, *J* = 11 Hz), 115.93–115.64 (C-3′, *J* = 22 Hz), 113.38 (C-7), 113.25 (C-4), 102.1–102.0 (C-10, *J* = 7.7 Hz), 23.83 (CH3). Elemental analysis calcd (%) for C_17_H_12_FNO_3_: C, 68.68; H, 4.07; N, 4.71; found C, 68.65; H, 4.01; N, 4.65. *m*/*z*: 297.08 (100.0%).


**(Z)-N-(2-(3-fluorobenzylidene)-3-oxo-2,3-dihydrobenzofuran-5-yl)acetamide (25):**


Yield: 82%; mp: 243.6 °C; ^1^H NMR (300 MHz, DMSO-d_6_): δ 10.22 (s, 1h, NH), 8.12 (d, 1H, *J* = 2.1 Hz, C-H_4_), 7.83–7.79 (m, 3H, C-H_6,2′,6′_), 7.59–7.51 (m, 2H, C-H_4′,7_), 7.31–7.39 (dt, 1H, *J* = 2.1, 8.4 Hz, C-H_5′_), 6.59 (s, 1H, C-H_10_), 2.07 (s, 3H, CH3). ^13^C NMR (75 MHz, DMSO-d_6_): δ 183.83(C-3), 168.6(CO), 163.85–160.62 (C-3′, *J* = 244 Hz), 161.48 (C-8), 147.37 (C-2), 135.77 (C-5), 134.3–134.19 (C-1′, *J* = 8.25 Hz), 131.08–130.97 (C-5′, *J* = 8.25 Hz), 129.05 (C-6), 127.62–127.58 (C-6′, *J* = 2.75 Hz), 120.53 (C-9), 117.45–117.15 (C-4′, *J* = 22.56 Hz), 117.02–116.73 (C-2′, *J* = 21.5 Hz), 113.53 (C-7), 113.25 (C-4), 110.69–110.66 (C-10, *J* = 2.75 Hz), 23.92 (CH3). Elemental analysis calcd (%) for C_17_H_12_FNO_3_: C, 68.68; H, 4.07; N, 4.71; found C, 68.58; H, 4.12; N, 4.73. *m*/*z*: 297.08 (100.0%).


**(Z)-5-amino-2-(2-fluorobenzylidene)benzofuran-3(2H)-one (26):**


Yield: 62%; mp: 161.3 °C; ^1^H NMR (300 MHz, DMSO-d_6_): δ 8.23 (dt, 1H, *J* = 1.65, 7.8 Hz, C-H_6′_), 7.51–7.47 (m, 1H, C-H_4′_), 7.37 (t, 1H, C-H_3′_), 7.34 (dt, 1H, C-H_5′_), 7.28–7.25 (d, 1H, *J* = 8.8 Hz, C-H_7_), 7.09–7.05 (dd, 1H, *J* = 2.5, 8.7 Hz, C-H_6_), 6.86 (d, 1H, *J* = 2.4 Hz, C-H_4_), 6.81 (s, 1H, C-H_10_), 5.28 (bs, 2H, NH_2_). ^13^C NMR (75 MHz, DMSO-d_6_): δ 184.05 (C-3), 164.75–161.71 (C-2′, *J* = 260 Hz), 158.19 (C-8), 148.04 (C-2), 145.83 (C-5), 131.84–131.73 (C-4′, *J* = 8 Hz), 131.2 (C-6′), 125.11–125.06 (C-5′, *J* = 3 Hz), 124.84 (C-6), 120.69 (C-9), 119.97–119.81 (C-1′, *J* = 12 Hz), 115.85–115.56 (C-3′), 113.22 (C-7), 105.54 (C-4), 100.82–100.72 (C-10).

Elemental analysis calcd (%) for C_15_H_10_FNO_2_: C, 70.58; H, 3.95; N, 5.49; found C, 70.44; H, 3.99; N, 5.32. *m*/*z*: 255.07 (100.0%).


**(Z)-5-amino-2-(3-fluorobenzylidene)benzofuran-3(2H)-one (27):**


Yield: 85%; mp: 159.7 °C; ^1^H NMR (300 MHz, DMSO-d_6_): δ 7.80–7.78 (m, 2H, C-H_2′,4′_), 7.57–7.50 (dt, 1H, C-H_5′_), 7.32–7.27 (m, 2H, C-H_7,6′_), 7.10–7.07 (dd, 1H, *J* = 2.5, 8.7 Hz, C-H_6_), 6.87 (d, 1H, *J* = 2.4 Hz, C-H_4_), 6.85 (s, 1H, C-H_10_). ^13^C NMR (75 MHz, DMSO-d_6_): δ 184.26(C-3), 163.81–160.58 (C-3′, *J* = 244 Hz), 158.49 (C-8), 147.63 (C-2), 145.01 (C-5), 134.58–134.47 (C-1′, *J* = 8.25 Hz), 130.85 (C-5′), 127.34 (C-6), 125.18 (C-6′), 120.84(C-9), 117.22–116.92 (C-4′, *J* = 22.5 Hz), 116.64–116.36 (C-2′), 113.38(C-7), 109.49–109.45 (C-10, *J* = 2.75 Hz), 106.08 (C-4). Elemental analysis calcd (%) for C_15_H_10_FNO_2_: C, 70.58; H, 3.95; N, 5.49; found C, 70.66; H, 4.08; N, 5.31. *m*/*z*: 255.07 (100.0%).


**(Z)-5-amino-2-(4-fluorobenzylidene)benzofuran-3(2H)-one (28):**


Yield: 82%; mp: 164.4 °C; ^1^H NMR (300 MHz, DMSO-d_6_): δ 8.09–8.04 (dd, 2H, *J* = 7.8 Hz, C-H_2′_), 7.81–7. 78 (d, 1H, C-H_4′_), 7.57–7.54 (d, 1H, *J* = 8.4 Hz, C-H_7_), 7.36 (t, 2H, C-H_3′_), 7.33 (d, 1H, C-H_4_), 6.98 (s, 1H, C-H_10_). ^13^C NMR (75 MHz, DMSO-d_6_): δ 183.5 (C-4), 165.39 (C-8), 164.38–161.07 (C-4′, *J* = 250 Hz), 145.91 (C-2), 137.62 (C-5), 133.73–133.62 (C-2′, *J* = 9 Hz), 128.55 (C-1′, *J* = 3 Hz), 124.24 (C-6), 123.95 (C-7), 120.81 (C-9), 116.26–115.97 (C-3′, *J* = 22 Hz), 113.14 (C-4), 111.04 (C-10). Elemental analysis calcd (%) for C_15_H_10_FNO_2_: C, 70.58; H, 3.95; N, 5.49; found C, 70.52; H, 4.07; N, 5.23. *m*/*z*: 255.06956 (100.0%).


**(Z)-N-(3-oxo-2-(3-(trifluoromethyl)benzylidene)-2,3-dihydrobenzofuran-5-yl)acetamide (29):**


Yield: 82%; mp: 252.1 °C; ^1^H NMR (300 MHz, DMSO-d_6_): δ 10.20 (bs, 1H, NH), 8.30–8.28 (m, 2H, C-H_2′,4′_), 8.13 (d, 1H, *J* = 2.1 Hz, C-H_4_), 7.85–7.81 (dd, 1H, *J* = 2.3, 8.8 Hz, C-H_6_), 7.79–7.72 (m, 2H, C-H_5′,6′_), 7.56–7.53 (d, 1H, *J* = 8.9 Hz, C-H_7_), 7.06 (s, 1H, C-H_10_), 2.07 (s, 3H, CH_3_). ^13^C NMR (75 MHz, DMSO-d_6_): δ 183.81 (C-3), 168.54 (CO), 161.44 (C-8), 147.58 (C-2), 135.82 (C-5), 134.74 (C-1′), 133.09 (C-6), 130.15 (C-6′), 129.94 (C-5′), 129.63–129.03 (C-3′, *J* = 31.7 Hz), 127.46 (C-4′), 126.17 (C-2′), 125.32–122.61 (CF3, *J* = 270 Hz), 120.48 (C-9), 113.5 (C-7), 113.23 (C-4), 110.18 (C-10), 23.91 (CH3). Elemental analysis calcd (%) for C_18_H_12_F_3_NO_3_: C, 62.25; H, 3.48; N, 4.03; found C, 62.09; H, 3.54; N, 3.98. *m*/*z*: 347.07693 (100.0%).


**(Z)-5-amino-2-(3-(trifluoromethyl)benzylidene)benzofuran-3(2H)-one (30):**


Yield: 82%; mp: >350 °C; ^1^H NMR (300 MHz, DMSO-d_6_): δ 8.31–8.29 (m, 2H, C-H_2′,4′_), 7.80–7.73 (m, 2H, C-H_-6′,5′_), 7.52–7.49 (d, 1H, *J* = 8.9 Hz, C-H_7_), 7.40–7.36 (dd, 1H, *J* = 2.3, 8.9 Hz, C-H_7_), 7.26 (s, 1H, C-H_4_), 7.05 (s, 1H, C-H_10_). ^13^C NMR (75 MHz, DMSO-d_6_): δ 183.7 (C-3), 161.09 (C-8), 147.65 (C-2), 134.75 (C-1′), 133.13 (C-5), 130.16 (C-6′), 130.01–129.59 (C-3′, *J* = 31 Hz), 128.44 (C-7), 127.48 (C-4′), 127.43 (C-2′), 126.11–122.78 (CF3, *J* = 250 Hz), 121.1 (C-9), 114.05 (C-6), 111.68 (C-4), 110.05 (C-10). Elemental analysis calcd (%) for C_16_H_10_F_3_NO_2_: C, 62.96; H, 3.30; N, 4.59; found C, 63.11; H, 3.35; N, 4.55. *m*/*z*: 305.06636 (100.0%).


**(Z)-4-((5-acetamido-3-oxobenzofuran-2(3H)-ylidene)methyl)benzoic acid (31):**


Yield: 71%; mp: 165.3 °C; ^1^H NMR (300 MHz, DMSO-d_6_): δ 10.28 (s, 1H, NH), 8.14 (d, 1H, *J* = 1.9 Hz, C-H_4_), 8.10–8.07 (d, 2H, *J* = 8.8 Hz, C-H_2′_), 8.05–8.02 (d, 2H, *J* = 8.8 Hz, C-H_3′_), 7.87–7.83 (dd, 1H, *J* = 2.2;8.8 Hz, C-H_6_), 7.55–7.53 (d, 1H, *J* = 8.89 Hz, C-H_7_), 6.98 (s, 1H, C-H_10_), 2.08 (s, 3H, NHCOCH_3_). ^13^C NMR (75 MHz, DMSO-d_6_): δ 183.85 (C-3), 168.54 (CO), 166.77 (COOH), 161.47 (C-8), 147.69 (C-2), 136.08 (C-1′), 135.82 (C-5), 131.35 (C-4′), 131.23 (C-3′), 129.74 (C-2′), 129.04 (C-6), 120.47 (C-9), 113.44 (C-7), 113.25 (C-4), 110.61 (C-10), 23.89 (CH3). Elemental analysis calcd (%) for C_18_H_13_NO_5_: C, 66.87; H, 4.05; N, 4.33; found C, 66.85; H, 4.12; N, 4.27. *m*/*z*: 323.07937 (100.0%).


**(Z)-N-(7-nitro-3-oxo-2-(3-phenoxybenzylidene)-2,3-dihydrobenzofuran-5-yl)acetamide (32):**


Yield: 91%; mp: 230.3 °C; ^1^H NMR (300 MHz, DMSO-*d*_6_): δ 10.51 (bs, 1H, NH), 8.71 (d, 1H, *J* = 2.2 Hz, C-H_6_), 8.35 (d, 1H, *J* = 2.2 Hz, C-H_4_), 7.89–7.86 (d, 2H, *J* = 7.8 Hz, C-H_2″_), 7.56 (t, 1H, *J* = 8.16 Hz, C-H_5′_), 7.42 (dt, 2H, C-H_3″_), 7.20–7.17 (d, 1H, *J* = 7.3 Hz, C-H_4′_), 7.14 (s, 1H, C-H_2′_), 7.09–7.06 (m, 2H, C-H_10, 6′_), 2.11 (s, 3H, CH_3_). ^13^C NMR (75 MHz, DMSO-*d*_6_): δ 178.66 (C-3), 169.01 (CO), 157.07 (C-1″), 156.36 (C-3′), 153.21 (C-8), 146.13 (C-2), 143.98 (C-7), 135.31 (C-5), 133.21 (C-1′), 130.64 (C-5′), 130.09 (C-3″), 127.01 (C-6), 124.60 (C-9), 123.69 (C-4″), 121.59 (C-6′), 120.86 (C-4), 119.76 (C-4′), 119.35 (C-2′), 118.66 (C-2″), 113.73 (C-10), 23.86 (CH3). Elemental analysis calcd (%) for C_23_H_16_N_2_O_6_: C, 66.34; H, 3.87; N, 6.73; found C, 66.21; H, 3.74; N, 6.71. *m*/*z*: 416.10084 (100.0%).


**(Z)-N-(2-(3-hydroxybenzylidene)-3-oxo-2,3-dihydrobenzofuran-5-yl)acetamide (33):**


Yield: 75%; mp: >350 °C; ^1^H NMR (300 MHz, DMSO-d_6_): δ 10.15 (s, 1H, NH), 9.67 (bs, 1H, OH), 8.11–8.10 (d, 1H, *J* = 2.2 Hz, C-H_4_), 7.83–7.80 (dd, 1H, *J* = 2.2;8.8 Hz, C-H_6_), 7.50–7.47 (d, 1H, *J* = 8.9 Hz, C-H_7_), 7.42 (d, 1H, C-H_2′_), 7.40–7.38 (d, 1H, *J* = 7.8 Hz, C-H_6′_), 7.30 (t, 1H, *J* = 7.8 Hz, C-H_5′_), 6.89–6.86 (dd, 1H, C-H_4′_), 6.82 (s, 1H, C-H_10_), 2.07 (s, 3H, C-H_3′_). ^13^C NMR (75 MHz, DMSO-d_6_): δ 183.65 (C-3), 168.41 (CO), 161.27 (C-8), 157.58 (C-3′), 146.69 (C-2), 135.54 (C-5), 132.9 (C-1′), 129.89 (C-5′), 128.8 (C-6), 122.64 (C-6′), 120.65 (C-9), 117.53 (C-3′), 117.46 (C-2′), 113.21 (C-7), 113.16 (C-4), 112.47 (C-10), 23.83 (CH3). Elemental analysis calcd (%) for C_17_H_13_NO_4_: C, 69.15; H, 4.44; N, 4.74; found C, 69.01; H, 4.48; N, 4.69. *m*/*z*: 295.08 (100.0%).


**(Z)-5-amino-2-(3-hydroxybenzylidene)benzofuran-3(2H)-one (34):**


Yield: 95%; mp: 225.2 °C; ^1^H NMR (300 MHz, DMSO-d_6_): δ 9.63 (bs, 1H, OH), 7.39 (d, 1H, C-H_2′_), 7.37–7.34 (d, 1H, *J* = 7.8 Hz, C-H_6′_), 7.28 (t, 1H, *J* = 7.8 Hz, C-H_5′_), 7.25–7.22 (d, *J* = 8.8 Hz, C-H_7_), 7.07–7.04 (dd, 1H, *J* = 2.2;8.5 Hz, C-H_4′_), 6.86–6.83 (m, 2H, C-H_4,6_), 6.70 (s, 1H, C-H_10_), 5.24 (bs, 2H, NH_2_). ^13^C NMR (75 MHz, DMSO-d_6_): δ 184.22 (C-3), 158.08 (C-8), 157.53 (C-3′), 146.99 (C-2), 145.57 (C-5), 133.21 (C-1′), 129.82 (C-5′), 124.66 (C-6), 122.42 (C-6′), 120.98 (C-9), 117.34 (C-3′), 117.1 (C-2′), 113.06 (C-7), 111.17 (C-4), 105.46 (C-10). Elemental analysis calcd (%) for C_15_H_11_NO_3_: C, 71.14; H, 4.38; N, 5.53; found C, 71.21; H, 4.34; N, 5.49. *m*/*z*: 253.07 (100.0%).

## 5. Conclusions

In the present study, 31 new aurone derivative compounds were synthetized. These new compounds were obtained by the substitution of the aurone scaffold at position 5 by amino and acetamido groups, and through various substitutions at the 2′, 3′ and 4′ positions. Antimicrobial testing identified two of these compounds, i.e., **10** and **20**, as the most active on both Gram-positive and -negative bacteria with MIC values as low as 0.78 µM. These were also the safest regarding human cells. The two selected compounds shared some structural similarity with the 5-acetamido substitution. The SAR study from this work correlates with the results previously obtained by Olleik et al. [14] showing that benzyloxy and isopropyloxy lead to interesting activities in aurone scaffolds with substitution on the A ring with amino or acetamido groups, improving the activity compared to the natural OH group. Taken together, these results confirm that the aurone scaffold is a promising structure that could be the starting point for the design of new antibacterial agents by diversifying the substitution pattern on A and B rings altogether.

## 6. Patents

Aurone derivatives and uses thereof for controlling bacteria and/or fungi. PCT/EP2021/069047. BOLLA Jean Michel., MARESCA Marc, NEULAT-RIPOLL Fabienne, OLLEIK Hamza, PERRIER-VIRET Josette, PIQUE Valérie. ROBIN Maxime.

## Figures and Tables

**Figure 1 antibiotics-13-00300-f001:**
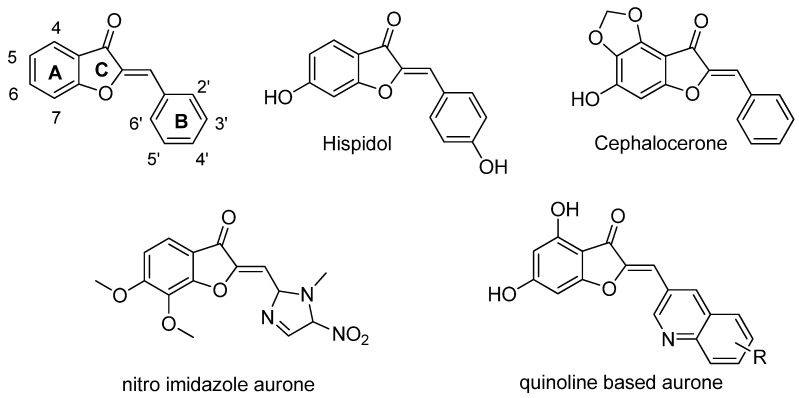
Structure of the aurone scaffold (top left), natural antimicrobial aurones, and examples of synthetic, heavily modified analogues.

**Figure 2 antibiotics-13-00300-f002:**
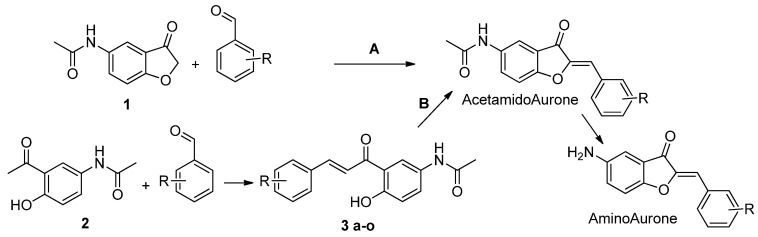
Synthetic route of the aurone derivatives. A: benzofuranone (**1**) 1 eq. and various benzaldehydes 1 eq. in Choline chloride/Urea (1/2), 80 °C, 2 h. B: (**2**) 1 eq. and various benzaldehydes 1 eq. in EtOH, LiOH 3 eq., 90 °C, 2 h. (**3 a-o**) 1 eq. and Mercuric acetate 1 eq. in pyridine, 110 °C, 1 h. Acetamido Aurone (**4**–**6**, **10**, **11**, **14**, **15**, **18**–**20**, **24**, **25**, **29**, **31**, **32**, **34**) are converted to their amino analogues (**7**–**9**, **12**, **13**, **16**, **17**, **21**–**23**, **26**–**28**, **30**, **33**) in EtOH, 0.5 M HCl, 100 °C, 2 h.

**Figure 3 antibiotics-13-00300-f003:**
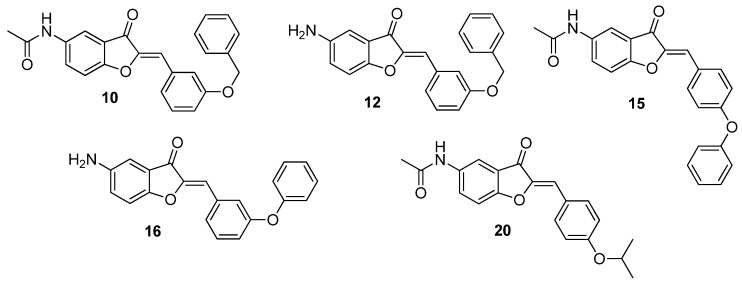
Structures of the more active aurone derivatives identified during the first screening (compounds **10**, **12**, **15**, **16,** and **20**).

**Figure 4 antibiotics-13-00300-f004:**
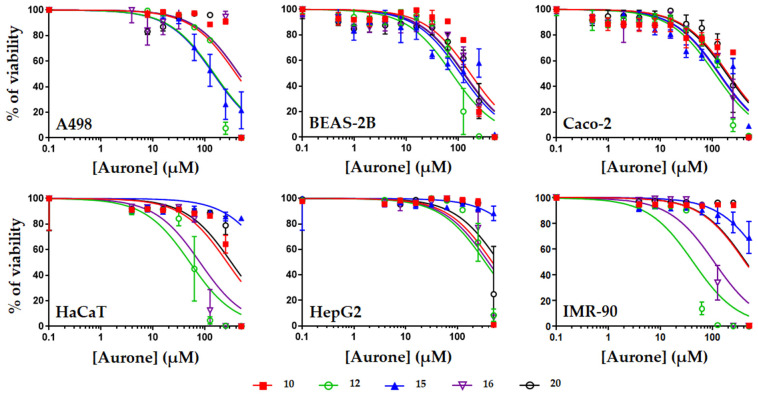
Evaluation of the toxicity of compounds **10**, **12**, **15**, **16**, and **20** on human cells. Human cells corresponding to kidney epithelial cells (A498), lung epithelial cells (BEAS-2B), intestinal epithelial cells (Caco-2), skin cells (HaCaT), liver cells (HepG2), or fibroblasts (IMR-90) were exposed to increasing concentrations of aurones for 48 h before measurement of the cell viability using resazurin. Results are expressed as % of cell viability, DMSO alone being used as negative control giving 100% viability. Data were plotted using GraphPad Prism 7 (means +/− S.D, *n* = 3).

**Figure 5 antibiotics-13-00300-f005:**
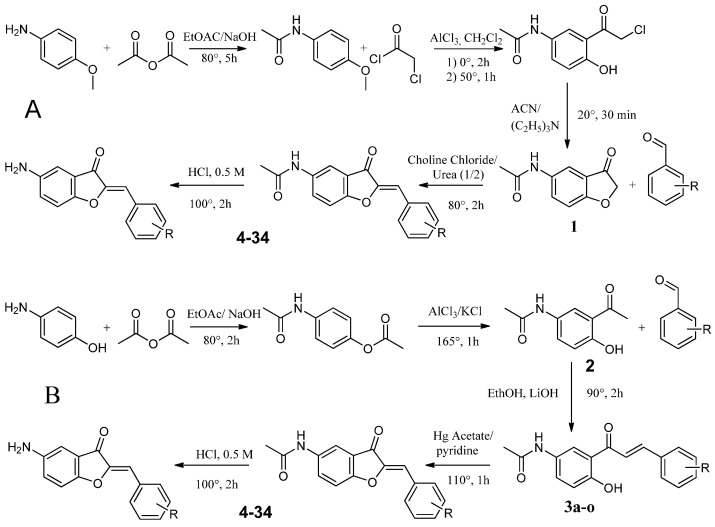
Route A and B used for the synthesis of 5-amino and 5-acetamido aurones.

**Table 1 antibiotics-13-00300-t001:** List of the synthetized aurones and their respective substitution on each position (OBn: Obenzyl, OPh: Ophenyl, OiPr: Oisopropyl).

Compound	5	2′	3′	4′
**4**	NHCOCH_3_	OCH_3_	H	H
**5**	NHCOCH_3_	H	OCH_3_	H
**6**	NHCOCH_3_	H	H	OCH_3_
**7**	NH_2_	OCH_3_	H	H
**8**	NH_2_	H	OCH_3_	H
**9**	NH_2_	H	H	OCH_3_
**10**	NHCOCH_3_	H	OBn	H
**11**	NHCOCH_3_	H	H	OBn
**12**	NH_2_	H	OBn	H
**13**	NH_2_	H	H	OBn
**14**	NHCOCH_3_	H	OPh	H
**15**	NHCOCH_3_	H	H	OPh
**16**	NH_2_	H	OPh	H
**17**	NH_2_	H	H	OPh
**18**	NHCOCH_3_	OiPr	H	H
**19**	NHCOCH_3_	H	OiPr	H
**20**	NHCOCH_3_	H	H	OiPr
**21**	NH_2_	OiPr	H	H
**22**	NH_2_	H	OiPr	H
**23**	NH_2_	H	H	OiPr
**24**	NHCOCH_3_	F	H	H
**25**	NHCOCH_3_	H	F	H
**26**	NH_2_	F	H	H
**27**	NH_2_	H	F	H
**28**	NH_2_	H	H	F
**29**	NHCOCH_3_	H	CF_3_	H
**30**	NH_2_	H	CF_3_	H
**31**	NHCOCH_3_	H	H	COOH
**32**	NHCOCH_3_	H	OH	H
**33**	NH_2_	H	OH	H
**34**	NHCOCH_3_	H	H	H

**Table 2 antibiotics-13-00300-t002:** Evaluation of the antimicrobial activities of the 31 newly synthetized aurone derivatives. The antimicrobial activities were determined using a MIC assay on species representative of Gram-positive bacteria (*B. subtilis*, *S. aureus*), Gram-negative bacteria (*E. coli*, *P. aeruginosa*), mycobacteria (*M. smegmatis*), and fungi (*C. albicans*). Amphotericin B and gemifloxacin were used as control antimicrobials for fungi and bacteria, respectively. The MIC values are given in µM (*n* = 2–3).

	Gram-Positive		Gram-Negative		Mycobacteria	Fungi
	*B. subtilis* ATCC6633	*S. aureus* ATCC6538P	*E. coli* ATCC8739	*P. aeruginosa* ATCC9027	*M. smegmatis* ATCC700084	*C. albicans* DSM10697
Gemifloxacin	0.03	0.06	0.06	0.25	2	-
Amphotericin B	-	-	-	-	-	0.62
**4**	50	100	>100	>100	>100	>100
**5**	>100	100	>100	>100	100	>100
**6**	>100	>100	>100	>100	>100	>100
**7**	>100	>100	>100	>100	>100	>100
**8**	>100	>100	>100	>100	>100	>100
**9**	>100	>100	>100	>100	>100	>100
**10**	3.12	12.5	12.5	25	50	50
**11**	>100	>100	>100	>100	>100	>100
**12**	25	100	>100	>100	50	>100
**13**	100	>100	>100	>100	>100	>100
**14**	>100	>100	>100	>100	>100	>100
**15**	50	100	>100	>100	50	>100
**16**	25	25	25	>100	50	>100
**17**	>100	>100	>100	>100	>100	>100
**18**	>100	>100	>100	>100	>100	>100
**19**	>100	>100	>100	25	>100	>100
**20**	25	12.5	25	>100	50	50
**21**	>100	>100	>100	>100	>100	>100
**22**	>100	>100	>100	>100	>100	>100
**23**	>100	>100	>100	>100	>100	>100
**24**	>100	>100	>100	>100	>100	>100
**25**	50	>100	>100	>100	100	>100
**26**	50	>100	>100	>100	>100	>100
**27**	>100	>100	>100	>100	>100	>100
**28**	>100	>100	>100	>100	>100	>100
**29**	>100	>100	>100	>100	>100	>100
**30**	>100	>100	>100	>100	>100	>100
**31**	50	100	>100	>100	100	>100
**32**	>100	>100	>100	>100	>100	>100
**33**	>100	>100	>100	>100	>100	>100
**34**	50	>100	>100	>100	>100	>100

**Table 3 antibiotics-13-00300-t003:** Determination of the cytotoxic concentrations of compounds **10**, **12**, **15**, **16**, and **20** on human cells. The cytotoxic concentrations 50 (CC_50_, in µM) (i.e., the concentrations of aurones causing 50% reduction of the cell viability after 48 h exposure) were calculated from Figure 3 using GraphPad Prism 7. Results are expressed as means +/− S.D (µM) (*n* = 3).

Compound	10	12	15	16	20
A498	398.2 +/− 164.6	152.4 +/− 31.4	145.7 +/− 17.3	452.3 +/− 147.9	453.0 +/− 46.4
BEAS-2B	169.0 +/− 28.3	74.6 +/− 12.1	109.5 +/− 17.4	129.6 +/− 18.3	125.9 +/− 17.2
Caco-2	199.6 +/− 33.5	111.7 +/− 15.0	136.8 +/− 7.2	131.7 +/− 18.0	186.5 +/− 27.8
HaCaT	268.5 +/− 51.6	51.3 +/− 9.5	>500	80.4 +/− 17.8	322.9 +/− 73.2
HepG2	472.4 +/− 145.9	343.4 +/− 68.0	>500	397.8 +/− 94.1	>500
IMR-90	421.4 +/− 119.3	42.4 +/− 9.6	>500	116.5 +/− 29.4	437.6 +/− 128
Mean CC_50_	321.5	129.3	130.6	218.0	305.1

**Table 4 antibiotics-13-00300-t004:** Determination of the therapeutic indexes of compounds **10**, **12**, **15**, **16**, and **20**. The therapeutic indexes (TI) of each aurone was calculated by dividing their CC_50_ on human cells (Table 3) by their MIC values on *S. aureus* from Table 2. MIC and CC_50_ are expressed in µM.

Compound	10	12	15	16	20
MIC on *S. aureus*	12.5	100	100	25	12.5
Lowest CC_50_	169.0	42.4	109.5	80.4	125.9
Highest CC_50_	472.4	343.4	>500	452.3	>500
Lowest TI	13.5	0.4	1.0	3.2	10.0
Highest TI	37.7	3.4	>5	18.0	>40

**Table 5 antibiotics-13-00300-t005:** Antimicrobial activities of compounds **10** and **20** on various bacterial and fungal species. The antimicrobial activities were determined using MIC assay as described in Section 4. The MIC values are given in µM (*n* = 2–3).

	10	20
**Gram-positive**		
*Bacillus anthracis* (CNR-charbon_04022)	12.5	6.25
*Bacillus cereus* (DSM31)	12.5	25
*Bacillus subtilis* (ATCC6633)	3.12	25
*Clostridium botulinum* (DSM1985)	0.78	3.12
*Clostridium difficile* (DSM1296)	12.5	3.12
*Clostridium perfringens* (ATCC13124)	>100	>100
*Enterococcus faecalis* (DSM2570)	50	100
*Enterococcus faecium* (DSM20477)	100	25
*Listeria monocytogenes* (DSM20600)	3.12	6.25
*Propionibacterium acnes* (ATCC6919)	100	>100
*Staphylococcus aureus* (ATCC6538P)	12.5	12.5
*MRSA* (ATCCBAA-1717)	12.5	25
*Streptococcus pyogenes* (DSM20565)	50	50
**Gram-negative**		
*Acinetobacter baumannii* (DSM30007)	12.5	25
*Brucella melitensis* (NR-256)	25	12.5
*Enterobacter cloacae* (DSM30054)	>100	>100
*Escherichia coli* (ATCC8739)	12.5	25
*Francisella tularensis* (NR-643)	50	12.5
*Helicobacter pylori* (ATCC43504)	12.5	12.5
*Klebsiella pneumonia* (DSM26371)	>100	>100
*Pseudomonas aeruginosa* (ATCC9027)	25	>100
*Salmonella enterica* (CIP80.39)	25	50
*Shigella flexneri* (ATCC12022)	25	50
*Vibrio alginolyticus* (DSM2171)	25	50
*Yersinia pestis* (NR-641)	12.5	12.5
**Mycobacteria**		
*Mycobacterium abscessus S* (CIP 104536^T^)	>100	>100
*Mycobacterium abscessus R*(CIP 104536^T^)	>100	>100
*Mycobacterium smegmatis* (ATCC700084)	50	50
*Mycobacterium tuberculosis H37Rv* (mc^2^6230)	>100	>100
**Filamentous fungi**		
*Aspergillus fumigatus* (DSM819)	>100	>100
*Fusarium oxysporum* (DSM62316)	25	25
**Yeasts**		
*Candida albicans* (DSM10697)	50	50
*Candida auris* (DSM21092)	50	12.5
*Candida glabrata* (DSM11226)	50	50
*Candida tropicalis* (DSM9419)	100	100
*Cryptococcus neoformans* (DSM11959)	25	25

**Table 6 antibiotics-13-00300-t006:** Safety evaluation for compound **10**. Therapeutic indexes (TI) values were calculated by dividing CC_50_ values by the lowest MIC value (obtained on *C. botulinum*, i.e., 0.78 µM) for compound **10** (Table 5).

	A498	BEAS-2B	Caco-2	HaCaT	HepG-2	IMR-90
CC_50_ (µM)	398.2	169.0	199.6	268.5	472.4	421.4
TI with MIC of 0.78 µM	510.5	216.6	255.8	344.2	605.6	540.2

**Table 7 antibiotics-13-00300-t007:** Safety evaluation for compound **20**. Therapeutic indexes (TI) values were calculated by dividing CC_50_ values by the lowest MIC value (obtained on *C. botulinum*, i.e., 3.12 µM) for compound **20** (Table 5).

	A498	BEAS-2B	Caco-2	HaCaT	HepG-2	IMR-90
IC_50_ (µM)	453.0	125.9	186.5	322.9	>500	437.6
TI with MIC of 3.12 µM	145.1	40.3	59.7	103.4	>160.2	140.2

## Data Availability

All data generated for this study are contained within the article.

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
