# Peer review of "Design and Synthesis of Novel Amino and Acetamidoaurones with Antimicrobial Activities"

_antibiotics, 2024, doi:10.3390/antibiotics13040300_

Round 1

Reviewer 1 Report

Comments and Suggestions for Authors

The authors presenting the antimicrobial activity of amino and acetamidoaurones, which showed activity against both gram positive and gram negative bacteria. The manuscript has introduced structural diversity, mainly replacing the phenolic groups in ring A with amine groups and has shown that some of the compounds retain antimicrobial activity. Below are a few of my observations:

1)        Scheme 1 has shown the acetamides to be compound 4-34. Infact the acetamides are hydrolyzed to amines to generate half of the compounds from 4-34. Therefore, the acetamide cannot be shown as representing compounds 4-34. The authors need to modify the scheme in such a way that there is not any ambiguity in labelling the compounds.

2)        Line 161: The more active is not proper language

3)        Line167-169: Sentence does not make sense without proper punctuation. Rephrase sentence.

4)        Line 170-172: I understand the urge to arrange the compounds in order of their antimicrobial activity. I am fine with the order but stating “with lowest MIC of 3.12, 12.5,25 and 50 mM” is not correct and should be refrained from. The different species of bacteria tested for these MICs cannot be jumbled up. The MICs stated should belong to the same species and strain to make sense. So either avoid writing the MIC values or if you would prefer, choose an average of the MICs for a few selected strains. 

5)        Lines 200-204 and Table 4: Calculating therapeutic index to showcase the effectiveness of a compound makes sense. Infact, if a compound has higher cell viability and lower MIC, its therapeutic index will be high. However, the way it has been done here I have some reservations. Table 4 shows the TI values of the compounds based on their lowest MIC values tested against different species of bacteria. As such these cannot be compared and having a table for it does not make logical sense. The TI values have to be accompanied by the bacterial species, without which it is irrelevant. 

6)        Line 233-236: Check sentence construction. Break down sentence if needed.

7)        Line 237: What is “on the opposite”?

8)        Section 4.2.1: Methods should be in past tense and passive voice. In 4.2.1.1-4 direct voice is used and then in 4.2.1.5 past tense and passive voice is used. The whole method should be written in the same passive voice as 4.2.1.5.

9)        Line 438, Section 4.2.1.5: It says conc. H2SO4 was used. However, in the scheme it shows HCl was used. Which is the correct one?

10)  Section 4.2.2: Change to past tense and passive voice.

11)  NMR data: For the 13C NMR data, remove all the carbon assignments. You can not say for sure which carbon is what, specially for aromatic carbons, unless you take the 2D NMR of each of one them. You can just include whether it is a CH, CH2 or CH3. For example: 168.41 (C), 128.71 (CH) and so on. 

12)  NMR spectras: Missing!! If NMR spectras were submitted, I did not find it in the attachments. NMR spectras of the reported compound must be attached to the supporting info. (MISSING DATA)

Comments on the Quality of English Language

Comments attached to the main comments section. Please go through the whole text and edit/rephrase as suitable. There are edits needed throughout the text.

Author Response

Dear Editor, Dear Reviewers,

               On behalf of all coauthors, we sincerely thank the reviewer for constructive criticisms and valuable comments, which were of great help in revising the manuscript. Accordingly, the revised manuscript has been systematically improved with new information and additional interpretations. Our responses to the referee’s comments are given below.

Comment : « Apart from the comments from
reviewers, we also detected high self-citation rate 9/25 (36%). Refs.
6,8,9,13,17,20,21,22,25 are from the co-authors ». 

Answer : Although we agree that the self-citation rate is quite high, it is due to the fact that to avoid detailed descriptions in the materials and methods leading to high autoplagiarism rate, we decided to cite previous assays descriptions already published by each partner of this study (many different lab, each of lab citing its own already described materials and methods. We hope the Editor will understand that i twill be hard/impossible to thus reduce the self-citation rate detected.

Reviewer 1 :

The authors presenting the antimicrobial activity of amino and acetamidoaurones, which showed activity against both gram positive and gram negative bacteria. The manuscript has introduced structural diversity, mainly replacing the phenolic groups in ring A with amine groups and has shown that some of the compounds retain antimicrobial activity. Below are a few of my observations:

1) Scheme 1 has shown the acetamides to be compound 4-34. Infact the acetamides are hydrolyzed to amines to generate half of the compounds from 4-34. Therefore, the acetamide cannot be shown as representing compounds 4-34. The authors need to modify the scheme in such a way that there is not any ambiguity in labelling the compounds.

Scheme 1. Synthetic route of the aurone derivatives. A: benzofuranone (1) 1 eq. and various benzaldehydes 1 eq in Choline chloride/Urea (1/2), 80°C, 2h. B: (2) 1 eq. and various benzaldehydes 1 eq. in EtOH, LiOH 3 eq., 90°C, 2h. (3 a-o) 1 eq. and Mercuric acetate 1 eq. in pyridine, 110°C, 1h. Acetamido Aurone (4-6,10,11,14,15,18-20,24,25,29,31,32,34) are converted to their amino analogues (7-9,12,13,16,17,21-23,26-28,30,33) in EtOH, 0.5 M HCl, 100°C, 2h.

Answer : We thanks the reviewer for the positive general comments and for the suggestion regarding scheme 1. Scheme 1 has been modified accordingly in the revised manuscript.

2)        Line 161: The more active is not proper language

Answer : We thanks the reviewer for the suggestion. This has been corrected in the revised manuscript.

3)        Line167-169: Sentence does not make sense without proper punctuation. Rephrase sentence.

Answer : We thanks the reviewer for the suggestion. This has been corrected in the revised manuscript.

4)        Line 170-172: I understand the urge to arrange the compounds in order of their antimicrobial activity. I am fine with the order but stating “with lowest MIC of 3.12, 12.5,25 and 50 mM” is not correct and should be refrained from. The different species of bacteria tested for these MICs cannot be jumbled up. The MICs stated should belong to the same species and strain to make sense. So either avoid writing the MIC values or if you would prefer, choose an average of the MICs for a few selected strains. 

Answer : We thanks the reviewer for the suggestion. Following the suggestion, we decided to avoid writing the MIC values as it was in the revised manuscript.

5)        Lines 200-204 and Table 4: Calculating therapeutic index to showcase the effectiveness of a compound makes sense. Infact, if a compound has higher cell viability and lower MIC, its therapeutic index will be high. However, the way it has been done here I have some reservations. Table 4 shows the TI values of the compounds based on their lowest MIC values tested against different species of bacteria. As such these cannot be compared and having a table for it does not make logical sense. The TI values have to be accompanied by the bacterial species, without which it is irrelevant. 

Answer : We thanks the reviewer for the suggestion. We understand the concern of the reviewer and agree with her/him. Accordingly, we changed the text and the Table 4 and indicated that TI were calculated using MIC obtained on S. aureus in the revised manuscript.

6)        Line 233-236: Check sentence construction. Break down sentence if needed.

Answer : We thanks the reviewer for the suggestion. This has been corrected in the revised manuscript.

7)        Line 237: What is “on the opposite”?

Answer : With all the respect due to the reviewer, except if we are wrong, we believe that « on the opposite » is correct and can be use (https://context.reverso.net/translation/english-french/on+the+opposite)

8)        Section 4.2.1: Methods should be in past tense and passive voice. In 4.2.1.1-4 direct voice is used and then in 4.2.1.5 past tense and passive voice is used. The whole method should be written in the same passive voice as 4.2.1.5.

Answer : We thanks the reviewer for the suggestion. This has been corrected in the revised manuscript.

9)        Line 438, Section 4.2.1.5: It says conc. H2SO4 was used. However, in the scheme it shows HCl was used. Which is the correct one?

Answer : We thanks the reviewer for the suggestion. This has been corrected in the revised manuscript (0.5 HCl, has been corrected).

10)  Section 4.2.2: Change to past tense and passive voice.

Answer : We thanks the reviewer for the suggestion. This has been corrected in the revised manuscript.

11)  NMR data: For the 13C NMR data, remove all the carbon assignments. You can not say for sure which carbon is what, specially for aromatic carbons, unless you take the 2D NMR of each of one them. You can just include whether it is a CH, CH2 or CH3. For example: 168.41 (C), 128.71 (CH) and so on. 

Answer : We thanks the reviewer for her/his comment.The attribution is done on previous data published by us (Casano, Gilles. (09.2010). Synthesis and complete assignment of the 1 H and 13 C NMR signals of new acetamido and aminoflavonoid derivatives. Magnetic resonance in chemistry. (48)9. p.738 - 744. DOI 10.1002/mrc.2638) we have made the 2D experiments on the starting aurone and the chemical shift are in agreement with the compare Flavone with only 0.1-0.5 ppm on 13C. The chemical environment of ring A and ring B is the same as the chalcone and flavone the only change is the Olefinic C10 of the aurone ring that appeared at 110 ppm corresponding of the Z form of the Aurone. And we have made series of 2’,3’,4’ substituant so we have all the data to confirm the complete assignment of aurone 13Cbased also with other publication.

12)  NMR spectras: Missing!! If NMR spectras were submitted, I did not find it in the attachments. NMR spectras of the reported compound must be attached to the supporting info. (MISSING DATA)

Answer : We are sorry for this missing file. This has been corrected in the revised manuscript.

Regards

Dr Marc Maresca

Reviewer 2 Report

Comments and Suggestions for Authors

antibiotics-2908642

This manuscript describes the synthesis and antimicrobial activity evaluation of a number of new aurone derivatives. Since the discovery of antimicrobial agents has growing importance nowadays, I believe that this work is of interest to the relevant scientists. The chemistry is not innovative but correctly performed and only a certain degree of antimicrobial activity was found for a limited number of the new compounds.

The substitution pattern provides some evidence for extracting SARs, but since only a small number of the presented derivatives possess a moderate (rather marginal) activity, it is not safe to extract conclusive results. Thus, in my opinion the Discussion part of the manuscript should not be so extended and I would encourage the authors to rephrase it and make it more concise.

Some other comments are listed below.

Line 42: world wide of 10 million…

Line 59: instead of ‘’ Despite some natural aurones”, better use “Even if some natural aurones”

In Lines 63-63, Instead of “Structurally, aurones are characterized by two 6-carbon rings (ring A and B) and a furanone-like third cycle (ring C).” I would propose a slightly different expression: Structurally, aurones are characterized by a benzofuran moiety bearing in position 2 a benzylidene type substituent.

Line 66: “or quinoline [16] groups”

Line 161 (and 175): the most active derivatives

Line 282: better use “31 new aurone derivatives were synthetized….”

Lines 285 and 816. There are no 5’-substitutions in this paper.

Line 298: the acetamido group is essential

Line 313: aurone 34

The description of the chemistry part in the experimental section (lines 412-434, 445-449) is preferable to be presented in past tense (like it is presented in lines 437-441, 451-465).

Melting points are provided for the target compounds. Are they recrystallized? In such a case recrystallization solvents should be mentioned.

Line 819: selected compounds

Comments on the Quality of English Language

Only minor editing is needed.

Author Response

Marseille, the 19th March 2024,

Dear Editor, Dear Reviewers,

               On behalf of all coauthors, we sincerely thank the reviewer for constructive criticisms and valuable comments, which were of great help in revising the manuscript. Accordingly, the revised manuscript has been systematically improved with new information and additional interpretations. Our responses to the referee’s comments are given below.

Reviewer 2 :

This manuscript describes the synthesis and antimicrobial activity evaluation of a number of new aurone derivatives. Since the discovery of antimicrobial agents has growing importance nowadays, I believe that this work is of interest to the relevant scientists. The chemistry is not innovative but correctly performed and only a certain degree of antimicrobial activity was found for a limited number of the new compounds.

Answer : We thanks the reviewer for her/his positive comments

The substitution pattern provides some evidence for extracting SARs, but since only a small number of the presented derivatives possess a moderate (rather marginal) activity, it is not safe to extract conclusive results. Thus, in my opinion the Discussion part of the manuscript should not be so extended and I would encourage the authors to rephrase it and make it more concise.

Answer : We thanks the reviewer for the suggestion. We tried our best to follow suggestions of the reviewer in the revised manuscript.

Some other comments are listed below.

Line 42: world wide of 10 million…

Answer : We thanks the reviewer for the suggestion. This has been corrected in the revised manuscript.

Line 59: instead of ‘’ Despite some natural aurones”, better use “Even if some natural aurones”

Answer : We thanks the reviewer for the suggestion This has been corrected in the revised manuscript to « Although some natural… »

In Lines 63-63, Instead of “Structurally, aurones are characterized by two 6-carbon rings (ring A and B) and a furanone-like third cycle (ring C).” I would propose a slightly different expression: Structurally, aurones are characterized by a benzofuran moiety bearing in position 2 a benzylidene type substituent.

Answer : We thanks the reviewer for the suggestion. This has been corrected in the revised manuscript.

Line 66: “or quinoline [16] groups”

Answer : We thanks the reviewer for the suggestion. This has been corrected in the revised manuscript.

Line 161 (and 175): the most active derivatives

Answer : We thanks the reviewer for the suggestion. This has been corrected in the revised manuscript.

Line 282: better use “31 new aurone derivatives were synthetized….”

Answer : We thanks the reviewer for the suggestion. This has been corrected in the revised manuscript.

Lines 285 and 816. There are no 5’-substitutions in this paper.

Answer : We thanks the reviewer for the suggestion. This has been corrected in the revised manuscript.

Line 298: the acetamido group is essential

Answer : We thanks the reviewer for the suggestion. This has been corrected in the revised manuscript.

Line 313: aurone 34

Answer : We thanks the reviewer for the suggestion. This has been corrected in the revised manuscript.

The description of the chemistry part in the experimental section (lines 412-434, 445-449) is preferable to be presented in past tense (like it is presented in lines 437-441, 451-465).

Answer : We thanks the reviewer for the suggestion. This has been corrected in the revised manuscript.

Melting points are provided for the target compounds. Are they recrystallized? In such a case recrystallization solvents should be mentioned.

Answer : We thanks the reviewer for her/his comment. Melting point are not made after recristalization and are uncorrected. A sentence have been include in 4.2 chemistry of the revised version.

Line 819: selected compounds

Answer : We thanks the reviewer for the suggestion. This has been corrected in the revised manuscript.

Regards

Dr Marc Maresca

Reviewer 3 Report

Comments and Suggestions for Authors

Please see attached comments.

Comments on the Quality of English Language

Please see attached comments.

Author Response

Marseille, the 19th March 2024,

Dear Editor, Dear Reviewers,

               On behalf of all coauthors, we sincerely thank the reviewer for constructive criticisms and valuable comments, which were of great help in revising the manuscript. Accordingly, the revised manuscript has been systematically improved with new information and additional interpretations. Our responses to the referee’s comments are given below.

Reviewer 3 :

The manuscript titled Design and synthesis of novel amino and acetamidoaurones with antimicrobial activities by Maresca, Robin and coworkers describes a synthesis of substituted aurone analogues and explores their antimicrobial activities from a wide range of bacteria and fungi. In my opinion, the manuscript was well written, and it meets the criteria of Antiboitics.

I would recommend for publication after the authors addressing the following notes. Detailed notes and comments are outlined below:

Answer : We thanks the reviewer for her/his positive feed-back.

  1. (Abstract) I suggest the authors make a connection between flavonoids and aurones. I would also suggest not mentioning the specific compounds like compounds 10, 12, 15, 16, and 20 as the abstract is a more general description of the manuscript.

Answer : We thanks the reviewer for this comment. The link between aurones and flavonoids has been added in the revised abstract. We believe that we need to keep the compounds numbers in the abstract in order to be able to comment on the activity and safety in this part of the manuscript. Not mentioning the compound numbers will make hard to understand the results for the reader. Aurones are a subclasses of the flavonoids. In the same way the antimicrobial activity have been explored for flavonol and flavone, we explored the antimicrobial activity of aurones. They come frome the same precursor, the chalcones.

  1. (Introduction) References are needed for many statements in this section, for example, ‘This is leading to an elevating mortality rate by infectious disease that can reach more than 10 billion of deaths by 2050 according to WHO.’

 Answer : We thanks the reviewer for this comment. New reference has been added regarding the WHO statement.

  1. (Line 68, Figure 1) The authors should move the mention of Figure 1 forward in the paragraph, as well as mention and display the C-ring in the general structure.

Answer : We thanks the reviewer for the suggestion. This has been corrected in the revised manuscript.

  1. (Scheme 1) What is the reasoning of choosing these two amide and amine groups over other functional groups and position 5 over positions 4, 6 and 7? Also, since the synthesized aurones have a huge conjugation structure, I have some concerns that positions 2’ and 3’ are not the same as 5’ and 6’. How can the authors tell the substitutions are always on the outer side of the molecule? Also, how can the authors tell the aurones are all Z-isomers instead of the E-configuration?

Answer : We thanks the reviewer for her/his comment.We have made te 5 acetamido and amino because it’s the first time these position is described with this kind of subtitution. The majority of the published work focuses on natural substitutions such as Hydroxy groups. No natural amino and acetamido substitued aurones have ever been described. We have built a complete pathway to obtain the desired aurones. About position 4 and 7, no description can be found and we choose to focused on the 5 position over position 6 in order to have a more focused SAR.. Regarding the 2’ an 3’ they are the same as 5’ and 6’ we could draw it on the opposite the ring C could turn around the C10-C1’ bond. It’s just a way of drawing and labelling for us. All the data of synthetic aurone described the Z isomer based on the 1H and 13C. The natural E isomer is rare and the 13C of the olefinic positon 10 falls between δ 119.8 and 122.2. For us the shift value of C-10 occurs at δ 112-114 ppm like other published results of Z isomer.

  1. (Tables 2-7) I suggest the authors add the unit (usually mM) in the table.

Answer : We thanks the reviewer for the suggestion. This has been corrected in the revised manuscript.

  1. (Discussion and Conclusion) The authors somehow included position 5’ in their description, which was not synthesized and discussed in the manuscript.

Answer : We thanks the reviewer for the suggestion. This has been corrected in the revised manuscript.

  1. (Discussion) The authors stated that ‘On the behalf of the results, it can be accepted that the acetamido groups is essential for the antibacterial capacity of the aurones.’ And I think this is too strong of a conclusion from the data. Also, the authors used the word interesting too much – maybe switch for words like intriguing or promising.

Answer : We thanks the reviewer for her/his comment. We have made and evaluated aurone without any substituent on the A ring and no antibacterial activity have been found (data not shown), but yes we could say seems to be essential.

  1. (Conclusion) I would suggest the authors not mention another group’s work in this section.

Answer : We thanks the reviewer for her/his comment. In fact, Olleik et al 2019 is a previous work of our group (ISM2 lab), so we believe we can mention our own work and compare it ot the present work.

  1. (General) The manuscript’s general format and language should be polished,

e.g.: • (Line 35) … antibacterial agents; (the authors used semicolons for the rest)…. • (Line 41) … according to World Health Organization (WHO). • (Line 64) (rings A and B) • (Line 66) e.g. • (Line 78) A: benzofuranone (1) 1 eq. and various benzaldehydes 1 eq. in choline chloride/urea (1/2), 80 °C, 2 h. • (Line 81) … aurone … • (Table 1, compound 13) NH2 • (Table 1) O-isopropyl or isopropyloxy or OiPr (same as ‘Obenzyl’ and ‘Ophenyl’) • (Line 162) i.e. • (Line 223) (n = 2-3) • (Line 291) … compounds 10 and 20 … • (Line 310) fluoro (i.e., 24-28) • (Line 409) Routes A and B … • (Line 414) (1.5 eq.) • (Section 4.2.1) A lot of C’s in ºC are missing • (Section 4.2.2) Z (configurations) and N (atoms) should be italicized. • (Compounds 18-21) m/z: m/z: (repetition) • (Line 821) …one of Olleik et al. [13] showing…

Answer : We thanks the reviewer for the suggestion. We tried our best to corrected all these mistakes in the revised manuscript.

  1. (References) Some references are missing the ending page (like 17, 18, 20-23). Also, some journal names are abbreviated while some are not - please keep them consistent

Answer : We thanks the reviewer for the suggestion. All journal name are now abbreviated, for the end pages, some journal such as Antibiotics and Toxins, doesn’t display it, citation 18 is a one page article

Regards

Dr Marc Maresca

We hope our review will be found suitable for Antibiotics.

Sincerely yours,

Dr Marc Maresca

Round 2

Reviewer 1 Report

Comments and Suggestions for Authors

I am glad that the authors took corrective actions on all my suggestions. For the NMR portion, I found the spectral and they look good. However, for the carbon assignment portion, it is okay for them to assign carbons as long as they put in some language referring to their previous publication from their lab based on which the carbons were assigned. I did not see any such obvious reference in the manuscript connecting the NMR data with their previous publication. Otherwise, everything else looks like is in order.